# Role of Ion-Exchange Resins in Hydrogenation Reactions

**Jordi H. Badia ***, **Rodrigo Soto**, **Eliana Ramírez**, **Roger Bringué**, **Carles Fité, Montserrat Iborra** **and Javier Tejero ***

Chemical Engineering and Analytical Chemistry Department, University of Barcelona, Martí i Franquès 1-11, 08028 Barcelona, Spain

* Correspondence: jhbadia@ub.edu (J.H.B.); jtejero@ub.edu (J.T.)

**Abstract:** The role of ion-exchange resins (IERs) as catalysts or catalysts supports, in hydrogenation reactions is revised and their potential application is presented. Both gel-type and macroreticular, basic or acid, IERs have been used for manifold metal-catalyzed hydrogenation processes in gas and liquid phase, including hydrogenation of alkenes, alkynes, carbonyls, arenes, nitroaromatics, and more. When available, qualitative relationships between the morphology and structure of resins and their performance as solid supports for metal catalysts are observed. Noble metals, such as Pt, Au, and Pd, and non-noble metals, such as Fe and Cu, have been introduced into IERs polymeric backbones by simple ion-exchange of a metal salt precursor with the resin, or by a combination of ion-exchange and other protocols, to obtain mono- and bimetallic catalysts supported on IERs. High yields towards target product, as well as the recyclability of metal-doped IERs, have been reported in the literature, with low metal leaching, which makes them highly interesting solid catalysts for a wide array of industrial applications. Multistep reaction processes, involving hydrogenation and hydration/cyclization/aldol condensation/etc., constitute promising applications due to the one-pot synthesis approach and relatively low temperatures required, which adds environmental interest in terms of process integration and optimization.

**Keywords:** ion-exchange resin; gel-type; macroreticular; hydrogenation; metal catalyst; bifunctional catalyst; one-pot process

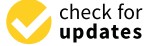



## 1. Introduction

To date, the role of ion-exchange resins (IERs) as catalysts for a number of industrial applications is a subject that has already been covered in great extension (e.g., [1–7]). Likewise, the role of IERs as support materials for metal-doped catalysts for hydrogenation reactions is not a new field either, since very complete reviews dating back more than twenty years ago can be found in the available literature, such as those by Corain and Králik (2000, 2001) [8,9]. However, scarce relevant works can be found in more recent times comprehensively dealing with the use of metal-doped IERs specifically as catalysts for hydrogenation reactions.

Noteworthy, a very complete review dealing with the use of polymeric materials as supports for a variety of reagents can be found in Dioos et al. (2006) [10]. In particular, Section 3.1 of that work offers an overview of a number of relevant aspects regarding the properties of the different types of polymeric backbones that can be used in this field. Regarding IER, a remarkable whole section (that is, Section 10 of that work) is devoted to the use of Nafion™ as support for metal catalysts, but little reference is made to metal-doped IERs of other types. Scarce mentions to the role of IERs as catalysts for several one-pot synthesis processes can be found in Climent et al. (2011, 2014) [11,12]. Furthermore, two reviews authored by Osazuwa and Abidin were published recently [13,14], where some works can be found cited therein regarding the use of IERs in different hydrogenations reactions, together with references related to the use of polymeric frameworks different from IERs as supports for metal catalysts (e.g., [15–18]). Table 1 lists some review papers

covering different aspects related to this field. At this point, it is worth mentioning that previous remarkable works (e.g., [19–23]) have not been explicitly included in the present review since, for the sake of brevity, it is limited to references dating back to the early 2000s (albeit a few exceptions have been made in Table 2).

**Table 1.** Previous review papers related to the use of IERs in hydrogenation reactions.

| Author(s) | Topic Reviewed | Date to Which the Literature is Covered |
|---|---|---|
| Corain and Králik [8] | Dispersion of metal nanoclusters inside IER | 2000 |
| Corain and Králik [9] | Generation of Pd nanoclusters inside IER | 2000 |
| Králik and Biffis [24] | Catalysis by metal nanoparticles supported on functional organic polymers | 2001 |
| Corain et al. [25] | IER as support for metal catalysts | 2003 |
| Gelbard [4] | IER applications as catalysts for organic syntheses | 2005 |
| Dioos et al. [10] | Immobilization of catalysts on polymeric supports | 2006 |
| Barbaro and Liguori [26] | Applications of IER as catalysts, including hydrogenations | 2008 |
| Corain et al. [27] | Synthesis and catalytic activity evaluation of metal nanoclusters inside IER | 2009 |
| Sarkar et al. [28] | Synthesis, characterization, and applications of polymer-supported metals | 2011 |
| Barbaro et al. [29] | Bifunctional metal/acid catalysts for one-pot processes involving hydrogenation and hydrolysis, lactonization, esterification, or cyclization, among others | 2012 |
| Climent et al. [12] | Conversion of biomass platform molecules into fuel additives and liquid hydrocarbon fuels | 2013 |
| Liguori et al. [30] | Metal nanoparticles immobilized on IER as catalysts for sustainable chemistry | 2015 |
| Osazuwa and Abidin [13] | Application of IER and other polymeric materials in hydrogenation reactions | 2017 |
| Osazuwa and Abidin [14] | Application of IER and other polymeric materials in esterification, transesterification, and hydrogenation reactions | 2019 |

Thus, the present work focuses on the use of IERs as catalysts, more precisely, as catalyst supports for hydrogenation reactions. A brief introduction is conducted in Section 1, which includes some generalities regarding IERs types and applications. Section 2 is focused on relevant morphological implications stemming from the introduction of metal nanoparticles into polymeric frameworks. Section 3, which is devoted to particular hydrogenation types applications, has been divided into several subsections, containing specific processes where IERs have been used as support for metal catalysts. Finally, overall concluding remarks and future scope are presented in Section 4.

*Ion-Exchange Resins: Main Types and Applications*

IERs are insoluble, organic materials consisting of a polymeric, amorphous backbone with a hydrophobic character with evenly distributed hydrophilic functional groups. Depending on their application, IER can present a wide variety of functional groups: acid (or cationic) resins usually contain sulfonic groups ($-SO_3H$), carboxylic groups ($-COOH$), or phenolic groups ($C_6H_4OH$) that can exchange cations in solution with $H^+$ ions, whereas basic (or anionic) resins can contain various amino groups ($-NH_3$, $=NH_2$, $=NH$) or quaternary substituted ammonium, being able to promote anions exchange with $OH^-$ ions [31]. IERs polymeric backbone, which is a crosslinked copolymer, consists of an irregular three-dimensional matrix of hydrocarbon chains. Typically, polymeric backbones can be based on perfluorosulfonic acid (i.e., Nafion™ resins), on polymethacrylate, or on polystyrene, among other monomers [3,13,32,33]. Typically, resins are thermally stable at temperatures up to 120–135 °C, but IERs exist with a higher thermal resistance that are able to operate at 170–190 °C (e.g., Amberlyst™ 45, Amberlyst™ DT, Purolite® CT482) [34,35].

Two main types of IERs are distinguished, based on their porosity: gel-type resins and macroreticular resins. Gel-type resins present a microporous, collapsed structure in a dry state that swells in contact with a polar solvent, such as water or certain alcohols, exhibiting thereby considerable porosity [36]. This type of resin can be obtained by free-radical crosslinking copolymerization of, for instance, styrene (ST) and divinylbenzene (DVB) monomers through suspension polymerization [33]. Macroreticular resins are composed of agglomerates of gel-type zones, with their respective polarity-sensitive microporous structure, that have a permanent macroporous structure in the dry state [33,36]. Macroreticular

resins can be synthesized by suspension polymerization of an ST-DVB monomer mixture in the presence of a diluent, which acts as a porogen that gives rise to the formation of a permanent porous structure within the highly crosslinked polymer particles upon its extraction. Functionalization of the resins is usually achieved by chemical modification of an already-formed copolymer matrix [23,33,36–38]. For instance, the acidic character of sulfonic IERs is usually achieved by direct contact with concentrated sulfuric acid between 303 and 413 K [36,39,40].

Among the wide range of IERs applications—including water softening and purification, demineralization, chromatography, pharmaceutical formulations, agricultural or mining, among others—their use as catalysts or catalyst supports has been well-known and appreciated for decades since they represent a conveniently cost-effective option for many industrial applications—e.g., alkylation, hydration, dehydration, etherification, esterification, condensation, cyclization, hydrolysis, oligomerization, C–C coupling, oxidation or hydrogenation reactions [3,13]. Some pivotal aspects that require careful evaluation when considering resins for a particular catalytic application include recyclability of the resin, economic and commercial effects of employing them, type of reactor, and reaction operating conditions, together with the already mentioned acid or basic nature of the functional groups and the polymer type and porosity [13].

## 2. Ion-Exchange Resins as Metal Catalysts Supports

Regarding the use of IERs as supports for metal catalysts, several works published in the early 2000s by Corain, Biffis, and coworkers need to be highlighted, since they offer a detailed overview assessing the morphological implications of introducing metal nanoparticles into the resins backbone [8,9,25,27,41–56]. Mostly, acrylic gel-type resins were used in those works, but macroreticular ST-DVB resins are also considered. For instance, Biffis et al. (2000a) and Corain et al. (2004) discussed the generation of size-controlled palladium nanoclusters inside gel-type resins [41,47], whereas the use of macroreticular resins is studied in D'Archivio et al. (2000) and Biffis et al. (2000b) [42,43]. Notably, the issue of metal nanoclusters size is comprehensibly discussed in Corain et al. (2008) [57].

Biffis and coworkers fixed Pd$^{II}$ to acrylic, mildly crosslinked, gel-type resins by contacting them with palladium acetate in a tetrahydrofurane:water solution, reaching metalation yields of about 90% [41]. Reduction of the metal was accomplished by exposure to sodium borohydride (NaBH$_4$) in ethanol. Through this procedure, they obtained 2–4 nm Pd crystallites that were homogeneously distributed in the resins. They concluded that the average size and size distribution of the generated Pd nanoclusters was independent of the metal content and of the crosslinking degree of the resins. The explanation given to their results was linked to the relatively low polymer chain density of the employed resins at which the nucleation and growth of the metal nanoclusters preferentially occur, giving rise to nanoparticles matching the dimensions of the nanometer-sized cavities inside the swollen polymer domains upon swelling [41]. From this, it follows that the resin-swollen polymer morphology can in fact act as a template to generate size-controlled metal nanoclusters suitable for the synthesis of resin-supported metal catalysts, since metal clusters would be blocked inside the largest available polymer cavities and, consequently, metal aggregates would not be able to continue growing. Such an approach was named Template Controlled Synthesis (TCS) approach (sketched in Figure 1), and it was confirmed by Corain et al. (2004) [47] using three independent morphological analyses: transmission electron microscopy (TEM), X-ray diffraction (XRD) and Inverse Steric Exclusion Chromatography (ISEC).

With respect to macroreticular resins, Biffis et al. (2000b), used commercial Lewatit® SPC 118 and UCP 118 to support Pd [42]. Textural features of the resins were determined by ISEC, revealing that the polymer mass was largely distributed in dense fractions. TEM of the obtained Pd/resins showed a heterogeneous distribution of Pd crystallites, appearing to be preferentially located either at the macropores surface or in the gel-type surface layer covering the polymer. Finally, CO chemisorption indicated that, in the dry state, Pd sites were practically inaccessible. From those results, the authors concluded that metalation

occurs both at the macropores surface and inside the highly crosslinked polymer mass, but while NaBH$_4$ reduction in the macropores surface is fast, it would be comparably slower inside the polymer denser fractions. Accordingly, some differences in Pd$^0$ concentration could exist within the resin which, ultimately, would act as a gradient to promote the migration of Pd$^{II}$ from the internal parts of the polymer towards the macropores surface by coalescence. The final consequence would be that the crystallites are located in well-defined gel-type domains at the macropore surface, which become accessible only in the swollen state [42]. A similar idea is briefly mentioned by Osazuwa and Abidin (2019), who simply stated that, depending on the pre-surface area and the bulk structure of the IER, the metal could migrate from the polymer to the pores after the reduction of the catalyst [13]. In any case, the fact that metal nanoparticles are heterogeneously distributed within macroreticular IERs compared to the homogeneous distributions observed in gel-type resins is a revealing finding to highlight.

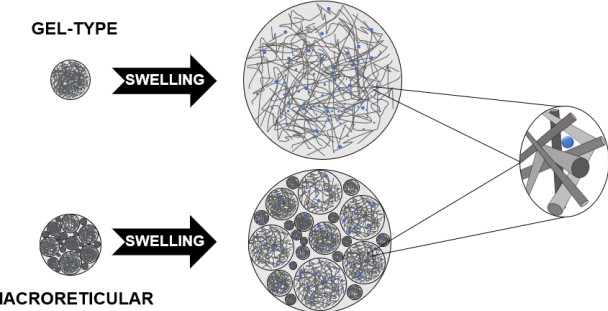

**Figure 1.** Schematic representation of the Template Controlled Synthesis (TCS) approach for a gel-type (top) and a macroreticular (bottom) IER. Inspired in [27,41,42].

The same commercial macroreticular resin (i.e., UCP 118, in different forms: without any metal, in both acidic and Na$^+$ forms, and with Pd in Na$^+$ form) was analyzed by electron spin resonance (ESR) and pulse field gradient spin echo-nuclear magnetic resonance (PGSE-NMR) techniques to assess their molecular accessibility to 2,2,6,6-tetramethyl-4-oxo-1-oxypiperidine (TEMPONE) dissolved in water in D'Archivio et al. (2000) [43]. The authors observed that both water and TEMPONE can penetrate inside the swollen resin, including the macro- and microporous domains, regardless of the Pd presence.

In a different study, Burato and coworkers used both gel-type and macroreticular IERs as supports for either Au or Pd nanoclusters and tested those catalysts in several reactions [48,51,52]. For instance, Pd$^{II}$ and Au$^{III}$ were linked onto an acrylic, moderately crosslinked, gel-type resin, obtaining well-sized-controlled Au$^0$ (2.2 nm) and Pd$^0$ (2.3 nm) nanoclusters, as expected according to their TCS approach (Figure 1). The Au$^0$/IER catalyst was tested in the oxidation of pentanal to pentanoic acid by molecular oxygen under mild conditions, revealing itself as twice as active as classic Au$^0$/C catalysts [51]. Later on, other acrylic resins were used following the same approach with success, and the catalytic tests revealed that the synthesized Au$^0$ catalyst supported on a very hydrophilic resin—i.e., poly-(4-vinylpyridine-acrylic acid-*N*,*N'*-methylenebisacrylamide)—was particularly promising for the selective oxidation of *n*-butanol to *n*-butanal [48]. The direct synthesis of hydrogen peroxide from molecular hydrogen and oxygen was tested with two acidic Pd$^0$/IER(H$^+$) and one non-acidic Pd$^0$-Au$^0$/IER catalysts, based on a commercial, macroreticular cation exchange resin (i.e., Lewatit$^®$ K2621/H$^+$) and two gel-type resins with –SO$_3$H and –SMe functional groups [52]. Interestingly, the Pd-Au nanoalloy supported on a hydrophilic resin prepared in this work was an effective catalyst for hydrogen peroxide synthesis.

In this line, Sterchele and coworkers prepared bimetallic Pd-Au (Pd: 1.0 wt.%; Au: 0.25–1.0 wt.%) and Pd-Pt (Pd: 1.0 wt.%; Pt: 0.1–1.0 wt.%) catalysts over commercial Lewatit$^®$ K2621 by simple ion-exchange in water and reduction with aqueous formaldehyde, which they tested in the direct synthesis of H$_2$O$_2$ [58]. Reported nanoparticle distributions varied depending on the catalyst composition, with average sizes of ca. 7 nm and 11 nm. Later on,

through X-ray absorption fine structure spectroscopy (EXAFS) under reaction conditions for the same synthesis, Centomo et al. (2015) observed that their Pd/K2621 catalyst was practically neither affected by the reaction medium (methanol) nor by the reaction mixture ($CO_2/H_2/O_2$), but that some of the PdO contained in the resin underwent reduction to metal Pd and some leaching in the presence of bromide ions [59].

The effect of adding Au to Pt/IER catalysts was studied by Sánchez and associates, who used Diaion$^{TM}$ WA30, a macroporous weak anion exchange resin with tertiary amine functional groups in an ST-DVB polymer matrix, as the support [60]. In previous work, they synthesized Pt/WA30 catalysts for the oxidation of glycerol to glyceric acid, showing high activity, selectivity, and stability [61]. Different preparation procedures were applied for the bimetallic catalysts, involving that either both Au and Pt were simultaneously exchanged with the resin (noted as PtAu samples), or that the exchange took place sequentially (and, in this case, Au-Pt catalyst denotes that Au was incorporated and reduced prior to incorporation of Pt; whereas, for Pt-Au catalysts, Pt was the first metal to be exchanged with the IER). In all cases, bimetallic catalysts formed an alloy, according to XPS results reported by the authors. Pt-Au catalysts showed similar behavior to monometallic Pt/WA30, but both PtAu and Au-Pt were more active. Regarding these two catalysts, Au-Pt samples were more active than PtAu samples with the same amount of metal, which was attributed to smaller metal nanoparticles (3.6 nm for Au-Pt in front of 5.4 nm) [60].

Pd (0.6–2.5 wt.%) and Pt (0.01–1.2 wt.%) were introduced in Nafion$^{TM}$-silica composites and Amberlyst$^{TM}$ 15, aiming at the direct hydroxylation of benzene with $O_2/H_2$ gas mixtures to produce phenol [62,63]. Amberlyst$^{TM}$ 15 is a commercial, strongly-acidic macroreticular IER with an ST-DVB matrix, functionalized with sulfonic groups. Nafion$^{TM}$ counts with an inert polymeric backbone and highly acidic sulfonic groups, with hydrophobic ($-CF_2CF_2-$) and hydrophilic ($-SO_3H$) regions. The influence of the solvent and the reduction protocol on the catalytic performance over the best catalyst (i.e., a Nafion$^{TM}$-based one) was evaluated, and the best results were obtained for a 3:1 water:methanol solvent and a reduced sample at 150 °C, yet little influence of the reduction conditions was observed [63]. The catalyst could be recycled after methanol washing followed by reduction, but catalytic activity could not be fully restored, which was attributed to the formation of heavy compounds [62].

In Bortolus et al. (2012), two hyper-crosslinked resins with permanent, accessible nanoporosity were prepared from a gel-type poly-chloromethylated styrene-divinylbenzene resin which, according to the authors, are promising supporting materials for metal catalysts [64]. The authors highlighted the importance of the solvent choice to be used in a catalyzed reaction, since the resin-solvent environment would strongly influence the mobility of the solute and, consequently, the effective reactivity. Jeřábek et al. (2013) investigated the same catalysts by elemental analysis, thermogravimetric analysis (TGA), SEM, X-ray microanalysis (XRMA), TEM, high-resolution TEM, selected area electron diffraction (SAED), nitrogen adsorption/desorption (BET analysis), ISEC, and specific absorption volume (SAV) measurements [65]. They evidenced that hyper-crosslinked resins present a permanent pore system comprising both micropores (<2.0 nm in diameter) and mesopores (2.2 nm), and that the metal nanoparticles had different sizes distribution all in the 1–6 nm range. Interestingly, they found that when formaldehyde was used as a reducing agent, more than 90% of the nanoparticles were less than 3 nm and that their radial distribution throughout the polymer beads was quite homogeneous [65].

More recently, an investigation on the effects of the support morphology for Pd/ST-DVB resins with different crosslinking degrees (that is, conventional macroreticular sulfonated ST-DVB resin and hyper-crosslinked sulfonated ST-DVB) can be found [66]. In their work, Frison and coworkers hypothesize that the very open pore structure of hyper-crosslinked resins would allow at the same time a fast inwards diffusion of the reagents and a fast outwards diffusion of the product in the direct synthesis of $H_2O_2$, making it a more suitable support than macroreticular resins, with its sulfonic groups located in the gel phase and mainly accessible upon swelling. The most favored hyper-crosslinked



Pd/ST-DVB resin in that work was more active than the tested Pd/C catalyst, despite its larger metal nanoparticles [66].

With regards to the effect that the resin functionality can exert on the preparation of metal-doped IERs, Moreno Marrodan and collaborators focused on the synthesis of Pd catalysts over two commercial gel-type resins (i.e., Dowex$^{TM}$ 50W × 2 and Dowex$^{TM}$ 1 × 2, that is a cationic resin with sulfonic groups and an anionic resin with trimethylbenzyl ammonium groups, respectively, both being ST-DVB resins with 2% crosslinking) [67]. By simply contacting the resins with aqueous solutions of palladium (II) nitrate (for the cationic resin) and potassium tetrachloropalladate (II) (for the anionic one) they obtained 60% metalation yields, reaching final Pd content of 1.1–5.1 wt.%. Three different approaches were tested regarding the activation of the catalysts: (1) reduction with aqueous NaBH$_4$, (2) reduction with H$_2$ gas, or (3) no pre-reduction of the catalyst at all, but, simply, allowing Pd$^{II}$ to undergo reduction to Pd$^0$ under reaction conditions. Observed Pd nanoparticles ranged 2.2–6.0 nm according to TEM, XRD, and small-angle X-ray scattering (SAXS), and reaction conversions as high as about 90% were obtained. Other factors investigated in that work were the solvent effect, IER bead size, and H$_2$ pressure. Regarding the solvent effect, it was seen that irrespectively of the substrate or the catalyst preparation method, the presence of water in the reaction solution invariably caused a decrease in catalyst activity, a loss of productivity upon reuse, and significant amounts of palladium leached, compared to pure methanol. Smaller bead sizes invariably yielded higher productivity, which entails resistance to internal mass transfer. With respect to H$_2$ pressure, conversion was higher at higher pressures (in the range of 1–8 bar), reaching a plateau, and selectivity towards fully hydrogenated species was promoted [67].

Concerning the catalytic activity for C–C coupling reactions, Van Vaerenbergh and colleagues studied Pd-doped IERs as catalysts for Suzuki cross-coupling reactions [68]. They prepared Pd-supported resins from the following commercial macroreticular resins: Ambersep® GT74 (ST-DVB with thiol (–SH) groups), Lewatit® K2629 (ST-DVB with sulfonic groups), and Lewatit® MP500OH (ST-DVB with quaternary amine (–CH$_2$N(CH$_3$)$_3$OH) groups). Results were compared to Amberlyst$^{TM}$ CH28, which is a commercial, macroreticular Pd-doped ST-DVB resin with sulfonic groups. They reported nanoparticles with average sizes of 1.34 nm (Ambersep® GT74, with a Pd load of 0.11 wt.%), 2.42 nm (Lewatit® K2629, Pd 0.08 wt.%), and 2.59 nm (Lewatit® MP500OH, Pd 0.11 wt.%), which contrast with the commercial Amberlyst$^{TM}$ CH28 (Pd ≥0.70 wt.%), with particles of ca. 7 nm, which the authors attribute to its higher metal loading. It was claimed that the metal size differences between the prepared catalysts are related to the stabilization strength of their functionalities, being stronger for smaller nanoparticles. Pd leaching, which was low for all catalysts, also seems to be related to the resin functionality, since the thiolic resin (Ambersep® GT74) yielded no leaching, Lewatit® MP500OH produced a 1.1% leaching, and Lewatit® K2629 showed a 4.8% leaching [68].

At this point, it is worth mentioning the work by Fujitsuka and coworkers [69]. Yet aiming at a different goal compared to other references contained in the present review (that is, they pursued the preparation of Pd/C, Pt/C, and Ni/C catalysts, after carbonization of metal-doped IER), high metal loadings (>10 wt.%) with small metal particle sizes (2.7–3.6 nm) were reported using commercial weakly basic and acidic IERs (i.e., Diaion$^{TM}$ WA30 and Diaion$^{TM}$ WK11, respectively). In their work, a very simple ion-exchange procedure is described, consisting in contacting small amounts of the resin (i.e., 5 g) with aqueous solutions of the designated metal salts for 24 h at room temperature with stirring, and adjusting the pH of the solution to 8.8 with aqueous NH$_3$. The metal nanoparticle sizes reported by Fujitsuka contrast with the larger 15–50 nm, and bigger ones, obtained in Prekob et al. (2020), which followed a very similar approach, carbonizing Pd-doped Amberlite$^{TM}$ IR-120 resin to obtain catalysts for gas-phase hydrogenation processes [70].

Table 2 lists some relevant aspects regarding the metal doping procedure of IER (such as the resin type, polymer composition, final metal content, and others) as retrieved from the cited references.



**Table 2.** Relevant information from the revised literature regarding the use of IERs as catalyst supports for hydrogenation reactions.

| Resin Type | Polymer Backbone | Crosslinking Agent | Metal | Precursor | Solvent | Metalation Yield (%) | Reduction Agent | Metal Nanoclusters Size (nm) | Ref. |
|---|---|---|---|---|---|---|---|---|---|
| Gel-type | Styrene, ST (92–96 mol%) | Divinylbenzene, DVB (4–8 mol%) | Ru (4 wt.%) | $Ru(NH_3)_6Cl_3$ | water | - | Sodium borohydride ($NaBH_4$) in ethanol (EtOH) | - | [21] |
| Gel-type | *N,N*-dimethylacrylamide, DMAA–styrene sodium sulphonate | *N,N′*-methylene (bis)acrylamide, MBAA (4 mol%) | Ru (4–8 wt.%) | $Ru(NH_3)_6Cl_3$ | water | - | $NaBH_4$ in EtOH | - | [21] |
| Gel-type | DMAA–potassium 1-methacryoyl ethylene 2-sulphonate | MBAA (4 mol%) | Ru (4–8 wt.%) | $Ru(NH_3)_6Cl_3$ | water | - | $NaBH_4$ in EtOH | - | [21] |
| Gel-type | DMAA (87–95 mol%) –2-sulfoethylmethacrylate, SEMA (4 mol%) | MBAA (1–9 mol%) | Pd | $Pd(OAc)_2$ | Tetrahydrofurane (THF):water (4:1 vol.) | From 86 to over 90 | $NaBH_4$ in EtOH | 2–4 | [41] |
| Macroreticular | Lewatit® SPC 118 (sulfonated polystyrene-divinylbenzene, ST-DVB, resin) | | Pd | $Pd(OAc)_2$ | - | - | - | - | [42] |
| Macroreticular | Lewatit® UCP 118 (ST-DVB resin) | | Pd (2 wt.%) | $Pd(OAc)_2$ | THF/water | | $NaBH_4$ in EtOH | - | [42] |
| Gel-type | Dodecyl methacrylate, DMA (46 mol%)–methyl methacrylate, MMA (46 mol%)–SEMA (4 mol%) | Ethylene dimethacrylate, EDMA (4 mol%) | Pd (1 wt.%) | $Pd(OAc)_2$ | THF | 90 | $H_2$ or $NaBH_4$ in EtOH/THF | - | [45] |
| Gel-type | DMA (92 mol%)–methacrylic acid, MAA (4 mol%) | EDMA (4 mol%) | Pd (1 wt.%) | $Pd(OAc)_2$ | THF | 100 | $H_2$ or $NaBH_4$ in EtOH/THF | - | [45] |
| Gel-type | DMA (92 mol%)–4-vinylpyridine, VP (4 mol%) | EDMA (4 mol%) | Pd (1 wt.%) | $Pd(OAc)_2$ | THF | 100 | $H_2$ or $NaBH_4$ in EtOH/THF | - | [45] |
| Gel-type | ST (92 mol%)–SEMA (4 mol%) | DVB (4 mol%) | Pd (1 wt.%) | $Pd(OAc)_2$ | THF | 100 | $H_2$ or $NaBH_4$ in EtOH/THF | - | [45] |
| Gel-type | DMA (92 mol%)–4-vinylpyridine (4 mol%) | Ethylene glycol dimethacrylate (4 mol%) | Pd (1 wt.%) | $Pd(OAc)_2$ | THF | 100 | $H_2$ or $NaBH_4$ in THF | 2.6–3.6 | [47] |
| Gel-type | DMAA (92 mol%)–SEMA (4 mol%) | MBAA (4 mol%) | Pd (1.67 wt.%) | $Pd(OAc)_2$ | Methanol (MeOH) | - | $NaBH_4$ in water | 3 | [71] |
| Macroreticular | Lewatit® K2641 (sulfonated ST-DVB resin) | | Pd (1.49 wt.%) | $Pd(OAc)_2$ | MeOH/acetone (ace) | 100 | - | - | [72] |
| Macroreticular | Lewatit® K2621 (sulfonated ST-DVB resin) | | Pd (1.34 wt.%) | $Pd(OAc)_2$ | MeOH/ace | 100 | - | - | [72] |
| Macroreticular | Lewatit® K2621 (sulfonated ST-DVB resin) | | Pd (1.37 wt.%) and Pt (0.14 wt.%) | $Pd(OAc)_2$ and $Pt(NH_3)_4(NO_3)_2$ | MeOH/ace | 100 | - | - | [72] |
| Gel-type | DMAA (92 mol%)–SEMA (4 mol%) | MBAA (4 mol%) | Fe (8.2 wt.%) and Pt (0.71 wt.) | $[Pt(NH3)_4]Cl_2$; $Fe(OAc)_2$ | water | 100 | $NaBH_4$ in water | | [50] |
| Gel-type | DMAA (92 mol%)–SEMA (4 mol%) | MBAA (4 mol%) | Zn (15.2 wt.%) and Pt (0.43 wt.) | $[Pt(NH3)_4]Cl_2$; $Zn(OAc)_2$ | water | 100 | $NaBH_4$ in water | | [50] |
| Gel-type | DMAA (92 mol%)–SEMA (4 mol%) | MBAA (4 mol%) | Co (14.1 wt.%) and Pt (0.89 wt.) | $[Pt(NH3)_4]Cl_2$; $Co(OAc)_2$ | water | 100 | $NaBH_4$ in water | 2–4 (Pt) | [50] |

**Table 2.** *Cont.*

| Resin Type | Polymer Backbone | Crosslinking Agent | Metal | Precursor | Solvent | Metalation Yield (%) | Reduction Agent | Metal Nanoclusters Size (nm) | Ref. |
|---|---|---|---|---|---|---|---|---|---|
| Gel-type | DMAA (88 mol%)–2-(methylthio)ethyl methacrylate, MTEMA (4 mol%) | MBAA (8 mol%) | Au (0.75 wt.) | $AuCl_3$ | Acetonitrile (ACN) | - | $NaBH_4$ in water | 2.2 | [48] |
| Gel-type | DMAA (92 mol%)–MTEMA (4 mol%) | MBAA (4 mol%) | Au (0.86 wt.) | $AuCl_3$ | ACN | - | $NaBH_4$ in water | 4.9 | [48] |
| Gel-type | DMAA (88 mol%)–MTEMA (4 mol%) | MBAA (8 mol%) | Pd (0.70 wt.%) | $Pd(OAc)_2$ | THF:water (2:1) | - | $NaBH_4$ in water | 2.3 | [48] |
| Gel-type | DMAA (86 mol%)–MTEMA (10 mol%) | MBAA (4 mol%) | Au (1.61 wt.) and Pd (2.43 wt.%) | $AuCl_3$ and $PdCl_2$ | water | - | $NaBH_4$ in water | - | [48] |
| Gel-type | DMAA (86 mol%)–poly-2-(methylthio)ethyl methacrylate (10 mol%) | MBAA (4 mol%) | Au (2.7 wt.) and/or Pd (0.8 wt.%) | $Pd(OAc)_2$ and/or $HAuCl4$ | ACN | 100 | $NaBH_4$ in water | - | [52] |
| Gel-type | DMAA (92 mol%)–2-methacryloyl ethanesulfonic acid (4 mol%) | DVB (4 mol%) | Pd (1.0 wt.%) | $Pd(OAc)_2$ | ace | 100 | EtOH | - | [52] |
| Macroreticular | Lewatit® K2621 (sulfonated ST-DVB resin) | | Pd (1.1 wt.%) | $Pd(OAc)_2$ | MeOH/ace | 100 | EtOH | 5 | [52] |
| Gel-type | DOWEX™ 50W × 2 (sulfonated, ST-DVB, with 2% nominal crosslinking), either in $H^+$ or lithiated ($Li^+$) form | | Pd (1.1–5.1 wt.%) | $Pd(NO_3)_2$ | water | 60 | $NaBH_4$ in water/gaseous $H_2$/in-situ under reaction conditions | 2.2–6.0 | [67] |
| Gel-type | DOWEX™ 1 × 2 (ST-DVB, with 2% nominal crosslinking, containing trimethylbenzyl ammonium groups) in chlorinated ($Cl^-$) form | | Pd (1.1 wt.%) | $K_2PdCl_4$ | water | 60 | $NaBH_4$ in water | - | [67] |
| Macroreticular | Lewatit® K2621 (sulfonated ST-DVB resin) | | Pd (1 wt.%) and Pt (0.1, 0.25, 0.5, 1 wt.%) | $[Pd(NH_3)_4]SO_4$; $[Pt(NH_3)_4](NO_3)$ | water | 100 | Formaldehyde (37%aq.), 3 h | - | [58] |
| Macroreticular | Lewatit® K2621 (sulfonated ST-DVB resin) | | Pd (1 wt.%) and Au (0.25, 0.5, 1 wt.%) | $[Pd(NH_3)_4]SO_4$; $[Au(en)_2]Cl_3$ | water | 100 | Formaldehyde (37%aq.), 3 h | - | [58] |
| Macroreticular | Diaion™ WA30 (ST-DVB resin with tertiary amine groups) | | Pt (1 wt.%) | $H_2PtCl_6$ | water | 100 | Hydrazine ($N_2H_4$) in the presence of NaOH, pH = 14 | 1.73–2.08 | [61] |
| Macroreticular | Diaion™ WA30 (ST-DVB resin with tertiary amine groups) | | Pt (0.4 wt.%) | $H_2PtCl_6$ | water | 100 | $N_2H_4$, pH = 14 | 1.9 | [60] |
| Macroreticular | Diaion™ WA30 (ST-DVB resin with tertiary amine groups) | | Au (0.4 wt.%) | $H_2AuCl_4$ | water | 100 | $N_2H_4$, pH = 14 | 1.9 | [60] |
| Macroreticular | Diaion™ WA30 (ST-DVB resin with tertiary amine groups) | | Pt and Au (0.4 wt.% total) | $H_2PtCl_6$ and $H_2AuCl_4$ | water | 100 | $N_2H_4$, pH = 14 | 3.6–5.4 | [60] |
| Macroreticular | Ambersep® GT74 (ST-DVB resin with thiol groups) | | Pd (0.11 wt.%) | $Pd(NO_3)_2 \cdot xH_2O$ | water | 100 | $NaBH_4$ in water | 1.34 | [68] |
| Macroreticular | Lewatit® K2629 (sulfonated ST-DVB resin) | | Pd (0.08 wt.%) | $Pd(NO_3)_2 \cdot xH_2O$ | water | 73 | $NaBH_4$ in water | 2.42 | [68] |
| Macroreticular | Lewatit® MP500OH (ST-DVB resin with quaternary amine groups) | | Pd (0.11 wt.%) | $K_2PdCl_4$ | water | 100 | $NaBH_4$ in water | 2.59 | [68] |

From Table 2, and to summarize the contents in Section 2, the following can be highlighted:

(1)   Doping IERs with metal catalysts is a well-known technique that can be roughly summarized as a two-steps procedure consisting of (a) metalation step, that is, dispersion of a metal into the IER, typically conducted by contacting the functionalized IER with a solution of a metal salt, which acts as a precursor, to allow ion-exchange $((MLn)^{2+}$ cations for cationic resins and $(MLn)^{2-}$ anions for anionic resins are required); and (b) activation of the catalyst, that is, reduction of the metal in its ionic form to its zero-valence form, to produce catalytically active metal nanoparticles. Scheme 1 illustrates such a technique for cation and anion exchange resins with an ST-DVB polymeric matrix.

(2)   Several metals have been used, including Pd, Pt, Au, and more. No works have been found reporting on particular difficulties in doping IERs with specific metals.

(3)   Several resin types have been used, including gel-type, macroreticular and hyper-crosslinked resins, either lab-made, commercially available or modified commercially available resins. Acrylic and ST-DVB resins stand out as the most widely used resins among the covered literature. The effect of a variety of functional groups has also been tested and, apparently, acidic resins would be more favored metal supports than basic ones.

(4)   Metal distribution within the resin beads depends on the resin type as follows: (a) homogeneous distributions are easily achieved in gel-type resins; (b) in contrast, when macroreticular resins are used, a clear heterogeneous distribution of the metal is observed, with the metal being concentrated at the macropores surface; and (c) as a strategy to circumvent this phenomenon, hyper-crosslinked resins can be used to obtain homogeneous metal distributions while maintaining a permanent, open mesopore structure of the polymeric backbone.

(5)   Several methods, solvents, and precursor metal salts have been used for the metalation step of IERs, with no clear evidence of a more favored procedure.

(6)   Metalation yields are generally high, easily achieving 100% anchoring levels of the metal contained in the precursor salt in the final metal-doped IER.

(7)   Several methods, solvents, and reducing agents are used in the reduction step of the metal ion to its zero-valence or elemental form. In this case, the specific catalytic application seems to be the determinant feature. Accordingly, metal-doped IERs aimed at catalyzing hydrogenation reactions do not appear to need a specific, separated reduction step, since the metal cation would be reduced in the reaction conditions. If this strategy is followed, notice that an induction period for the catalyst would take place before the reaction which must be taken into account for further results interpretation.

(8)   A variety of metal nanoparticle size distributions is reported in the literature, with values as low as about 1 nm, and a remarkably good size control can be achieved.

Cation exchange resins: X=SO$_3$, CO$_2$ || Y= H, Li, Na          Anion exchange resins: X= Ni(CH$_3$)$_3$, NH$_3$ || Y= Cl, OH

**Scheme 1.** Representation of the metal nanoparticles immobilization pathway on cation (red) and anion (blue) exchange resins with a PS-DVB polymeric matrix. Inspired in [30].

## 3. Ion-Exchange Resins Application to Hydrogenation Reactions

Hydrogenation can be defined as the incorporation of hydrogen atoms, either from gaseous H$_2$ or from other sources, to unsaturated compounds, that is compounds with

double (C=C) or triple (C≡C) bonds, including those with carbonyl (C=O) or nitrile (C≡N) groups.

The application of metal-doped IERs in hydrogenation reactions has been the focus of continuous attention for the research community with many relevant works published in the field. For instance, only in Corain and Králik (2000), the following hydrogenation reactions are mentioned: methyl isobutyl ketone (MIBK) production; hydrogenation of diolefins, acetylenes and carbonyl compounds from C4 hydrocarbon streams; etherification–hydrogenation of mixtures of unsaturated hydrocarbons to give blends of alkanes and branched ethers for the manufacture of unleaded petrol (i.e., BP Etherol process [73,74]); $O_2$ removal from industrial waters; hydrogenation of cyclohexene; hydrogenation of nitroaromatics; hydrogenation of 2-ethylanthraquinone; and hydrogenation of nitrate and nitrite ions to ammonia [8]. In their work, Corain and Králik considered applications at temperatures ranging from room temperature to 120 °C, macroreticular and gel-type resins, and metals such as Pd, Ni, Cu, or Pt. They also provided details regarding both the dispersion of the metal in the IERs (i.e., metalation step) and the activation protocol (i.e., reduction step) of the metal-doped IERs, which have already been included to some extent in Section 2 of the present work.

Further work can be found with respect to some of the quoted applications, such as cyclohexene hydrogenation into cyclohexane (e.g., [44]), hydrogenation of alkynes such as 2-butyne-1,4-diol and phenylacetylene (e.g., [75]), or hydrogenation of 2-ethylanthraquinone to 2-ethylanthrahydroquinone (e.g., [45]). Other studies deal specifically with multistep reaction processes, such as the mentioned MIBK or, for instance, $\gamma$-valerolactone (GVL) production (e.g., [12,29,76,77]), involving catalytic activity not only for hydrogenation because of the metal sites, but also for at least another reaction typically requiring acid/basic sites (e.g., hydrolysis, lactonization, esterification, cyclization, or condensation). Therefore, metal-doped IERs for such applications must present a bifunctional nature, incorporating active sites of different characteristics on the same support. Regarding reaction conditions, pressures ranging from 1–50 bar and temperatures as high as 150 °C can be found, depending on the application and IER used.

A general overview suggests that the range of potential applications of metal-doped IERs in hydrogenation reactions is notoriously wide, as is the range of experimental conditions specifically required for a particular process. Aiming to scrutinize such scenarios and for the sake of clarity, the next sections have been divided into the following: (1) alkenes hydrogenation; (2) alkynes hydrogenation; (3) carbonyl hydrogenation; (4) substituted arenes hydrogenation; (5) hydrogenation of nitroaromatic compounds to their corresponding amines; (6) reduction of nitrites and nitrates; (7) one-pot, multistep syntheses involving hydrogenation, together with other reactions (e.g., esterification, cyclization, condensation); and (8) other related reactions, including the direct synthesis of hydrogen peroxide from molecular oxygen.

### 3.1. Hydrogenation of Alkenes

Four Pd-doped (0.25–0.45 wt.%), sulfonated, synthetic IERs were tested in the liquid-phase hydrogenation of cyclohexene to cyclohexane and cyclohexen-1-one to cyclohexanone by Zecca and coworkers [44]. All resins were exchanged with palladium acetate, and $Pd^{II}$ was reduced to $Pd^0$ with $NaBH_4$ in ethanol. Depending on their composition, the solvent affinities of the resins spanned from moderately hydrophobic to clearly hydrophilic—the lower the ST amount in the resin, the higher its hydrophilicity. No effect of the hydrophobic/hydrophilic character of the resin was observed on the reported catalytic activity. Interestingly, resins with more nitrogen atoms in their structures (i.e., those based on *N,N'*-methylene(bis)acrylamide and *N,N*-dimethylacrylamide) were found to be the most active ones, with rates increasing linearly with the nitrogen/Pd molar ratio, and no metal leaching was observed in any case.

The role of the gel-type IERs properties in the hydrogenation of the C=C bonds of maleic and fumaric acids, viz. the *cis* and *trans* isomers of butenedioic acid, was investigated

in Drelinkiewicz et al. (2008) [78]. They synthesized Pd-doped (0.25–2.0 wt.%, 2–3 nm) acrylic resins with crosslinking degrees of 3 and 10%. The observed swelling of the polymer mass during the hydrogenation was linked to the catalytic activity: the same hydrogenation rates were observed for both maleic and fumaric acids in the absence of swelling, while the hydrogenation rate of maleic acid was much higher than that of fumaric acid in the swollen state. Interestingly, a "bell-like" dependence between initial rates and metal content was found, with maximum activity at 0.5–1.0 wt.% Pd content.

Among the reactions tested in Moreno Marrodan et al. (2012), hydrogenation of the C=C bond in methyl 2-acetamidoacrylate and in *trans*-4-phenyl-3-buten-2-one was studied over Pd-doped, gel-type, cationic and anionic resins [67]. Reported conversions were 91.7% and 98.2%, respectively, with selectivities of 100% and 83.7%, over lithiated Pd/Dowex$^{TM}$ 50W × 2 resin without any separated pre-reduction step of the ionic Pd$^{II}$ to Pd$^0$. Remarkably, neither activity loss upon recycling of the catalyst nor Pd leaching was observed and selectivity was unaffected after 6 cycles using methanol as the reaction solvent.

Madureira and associates hydrogenated fatty acid methyl esters (i.e., methyl undecenoate, methyl oleate, methyl ricinoleate, and methyl linoleate) and vegetable oil-based compounds (i.e., glycerol trioleate, castor oil, jojoba oil, olive oil, and very high-oleic sunflower oil) over Ru/Dowex$^{TM}$ 50W × 2 under mild conditions (30 °C and 10 bar H$_2$) [79]. Regarding the metal impregnation procedure (consisting of ion exchange with an aqueous metal salt solution followed by NaBH$_4$ reduction), the authors indicate that the resin must be in the Na$^+$ form to improve metalation yield and to avoid Ru leaching due to the presence of acidic species. According to their results, the catalytic activity of the catalysts was higher for shorter hydrocarbon chain length: hydrogenation of methyl undecenoate (C$_{12}$) was faster than methyl linoleate (C$_{19}$), showing 100% conversion in 60 min in front of about 50% conversion in 210 min, respectively. Moreover, the double bond in the larger unsaturated esters was located in an internal position, which hindered the reaction rate [79].

In a different approach, a lithiated Dowex$^{TM}$ 50W × 2 resin was used to immobilize phosphine–rhodium complexes to catalyze the asymmetric hydrogenation of prochiral olefins, such as methyl 2-acetamidoacrylate, under mild conditions (room temperature, under 1–5 bar H$_2$) [80–82]. Reported yields mount up to 99.9%, even in the second recycle, with a detected Rh leaching of 2.0% [80].

Hydrogenation of cyclohexene in supercritical CO$_2$ over metal-doped Amberlyst$^{TM}$ 15 was studied by Seki and associates [83]. The metals employed (Pd, Pt, Ru, and Rh) were incorporated into Amberlyst$^{TM}$ 15 in different amounts from 0.5 to 2 wt.% to obtain monometallic catalysts. A bimetallic Pd-Pt catalyst was also tested. The highest conversions (i.e., up to 99%) and selectivity towards cyclohexane (i.e., >99%) were reported for Pd-based catalysts. Results obtained with Pt, Ru, and Rh catalysts were clearly less favorable with, conversion values in the range of 1–4%.

Some relevant hydrogenation reactions of alkenes found in the covered literature are shown in Scheme 2. Table 3 lists resin types, tested reactions, and obtained results of some of the reviewed works.

As Scheme 2 and Table 3 reveal, Pd is the preferred metal to hydrogenate C=C bonds, as described in previous works [85,86], but Ru-based catalysts have also been used to hydrogenate stronger C=C bonds, such as the conjugated ones in benzene rings. Regarding the IERs, both synthetic and commercial resins have been used, but mostly sulfonated and gel-type ones. Noteworthy, the anion exchange resin tested by Moreno Marrodan and coworkers yielded no significant catalytic activity in several hydrogenation reactions [67]. Different strategies regarding the activation step of the metal catalyst emerge from the revised literature, including chemical reduction of the ionic metal with NaBH$_4$ and in situ reduction during reaction conditions (with no separate pre-reduction step). Depending on the author and explored reaction, the reported results differ substantially. For instance, Jeřábek reported much higher conversions when pre-reducing the catalyst at 1 MPa H$_2$ in water at 30 °C for 2 h compared to in situ reduction during cyclopentene hydrogenation to

cyclopentane (Table 3), whereas Moreno Marrodan reported the opposite behavior in the hydrogenation reaction of methyl 2-acetamidoacrylate to methyl acetylalaninate [67,84].

**Scheme 2.** Examples of hydrogenation reactions of alkenes over metal-doped IER in the revised literature.

### 3.2. Hydrogenation of Alkynes

In Drelinkiewicz et al. (2009), self-made gel-type resins functionalized by amine were used, which were obtained by suspension polymerization of a mixture of glycidyl methacrylate, ST, and diethylene glycol dimethacrylate (DEGDMA). As metal nanoparticles, Pd (0.125–0.50 wt.%) of considerably large sizes (30–50 nm) were obtained when Pd was incorporated on dry resin beads using an aqueous solution of $PdCl_2$ as the precursor, which contrasts with the nano-scaled Pd particles obtained when $Pd(OAc)_2$ was used. However, it was the 30–50 nm Pd catalyst the one exhibiting better results for the hydrogenations of 2-butyne-1,4-diol and phenylacetylene to their respective alkenes, with selectivity values ca. 92–95% at 90% conversion of the alkyne. The better performance of the large Pd nanoparticles-containing IER is related to its metal distribution (i.e., mostly, in the outer shell of the polymer beads, forming a thin layer) [75].

**Table 3.** Hydrogenation reactions of alkenes over metal-doped IERs in the literature.

| Resin | | | Metal | Reduction Protocol | Tested Reaction | Reaction Conditions | Conversion (%) | Selectivity (%) | Ref. |
|---|---|---|---|---|---|---|---|---|---|
| Type | Polymer | Functional Group | | | | | | | |
| Macroreticular | Lewatit® OC 1038, ST-DVB | –SO$_3$H | Pd (0.3 wt.%) | In situ | Hydrogenation of cyclopentene | 2 mL cyclopentene, T = 30 °C; P = 1 MPa H$_2$; t = 2 h | 30 | - | [84] |
| | | | | 1 MPa H$_2$ in cyclopentene at 30 °C for 2 h | | | 28.7 | - | [84] |
| | | | | 1 MPa H$_2$ in water at 30 °C for 2 h | | | 100 | - | [84] |
| | | | | 40% aq. formaldehyde at 50 °C for 1 h | | | 77.8 | - | [84] |
| Gel-type | ST-DVB (4 mol%) | –SO$_3$H | Ru (4 wt.%) | NaBH$_4$ in EtOH | Partial hydrogenation of benzene to cyclohexene | 2 mL benzene, 0.75 mL water; T = 100 °C; P = 1.5 MPa | 43.1 | 3.0 | [21] |
| | ST-DVB (8 mol%) | | Ru (4 wt.%) | | | | 44.6 | 4.4 | [21] |
| | DMAA–styrene sodium sulphonate–MBAA | | Ru (4 wt.%) | | | | 45.4 | 5.6 | [21] |
| | | | Ru (8 wt.%) | | | | 45.3 | 5.4 | [21] |
| | DMAA–potassium 1-methacryoyl ethylene 2-sulphonate–MBAA | | Ru (4 wt.%) | | | | 47.2 | 8.1 | [21] |
| | | | Ru (8 wt.%) | | | | 42.3 | 7.0 | [21] |
| Gel-type | 2-Hydroxyethyl methacrylate (HEMA), ST, diethylene glycol dimethacrylate (DEGDMA, 3–10 mol%) | –COOH and C=O | Pd (0.25–2.0 wt.%) | NaBH$_4$ in THF:MeOH (9:1) | Hydrogenation of maleic/fumaric acid | C$_{substrate}$ = 0.043 M in THF; T = 22 °C; atmospheric pressure | - | 100 | [78] |
| Macroreticular | Amberlyst™ 15, ST-DVB | –SO3H | Pd (0.5 wt.%) | 100 °C for 1 h under H$_2$/N$_2$ flow | Hydrogenation of cyclohexene | Supercritical CO$_2$, T = 60 °C, total P = 160 bar, cyclohexene:H$_2$ = 1:1.8 | 86 | 95 | [83] |
| | | | Pd (1 wt.%) | | | | 99 | >99 | [83] |
| | | | Pd (2 wt.%) | | | | 56 | 99 | [83] |
| | | | Pt (1 wt.%) | | | | 2 | 52 | [83] |
| | | | Pd (0.5 wt.%), Pt (0.5 wt.%) | | | | 59 | 96 | [83] |
| | | | Ru (1 wt.%) | | | | 4 | 94 | [83] |
| | | | Rh (1 wt.%) | | | | 1 | 24 | [83] |

**Table 3.** *Cont.*

| Resin | | | Metal | Reduction Protocol | Tested Reaction | Reaction Conditions | Conversion (%) | Selectivity (%) | Ref. |
|---|---|---|---|---|---|---|---|---|---|
| Type | Polymer | Functional Group | | | | | | | |
| Gel-type | Dowex$^{TM}$ 1 × 2, ST-DVB | –C10H16N(Cl) | Pd (1.1 wt.%) | NaBH$_4$ in water | Several C=C, C≡C, and C=O hydrogenation reactions | C$_{substrate}$ = 0.17 M in MeOH, room temperature, P = 0.8 bar H$_2$, 20 min | no observed catalytic activity | | [67] |
| Gel-type | Dowex$^{TM}$ 1 × 2, ST-DVB | –SO3(Li) | Pd (1.2 wt.%) | NaBH$_4$ in water | Hydrogenation of methyl 2-acetamidoacrylate | C$_{substrate}$ = 0.17 M in MeOH, room temperature, P = 0.8 bar H$_2$, 20 min | 56.3 | 100 | [67] |
| | | –SO3(Li) | Pd (1.2 wt.%) | 2 bar H$_2$ in MeOH | | | 87.4 | 100 | [67] |
| | | –SO3(Li) | Pd (1.3 wt.%) | In situ | | | 91.7 | 100 | [67] |
| Gel-type | Dowex$^{TM}$ 1 × 2, ST-DVB | –SO3(Li) | Pd (1.3 wt.%) | In situ | Hydrogenation of trans-4-phenyl-3-buten-2-one | C$_{substrate}$ = 0.17 M in MeOH, room temperature, P = 0.8 bar H$_2$, 20 min | 98.2 | 83.7 | [67] |
| Gel-type | Dowex$^{TM}$ 1 × 2, ST-DVB | –SO3(Na) | Ru (0.9 wt.%) | NaBH$_4$ in water | Hydrogenation of methyl undecenoate, methyl oleate, methyl ricinoleate, or methyl linoleate | n$_{substrate}$ = 1.6 mmol in water (0.25 mL) and n-heptane (6 mL), T = 30 °C, P = 10 bar H$_2$, t = 60–210 min | 50–100 | - | [79] |
| Gel-type | Dowex$^{TM}$ 1 × 2, ST-DVB | –SO3(Na) | Ru (0.9 wt.%) | NaBH$_4$ in water | Hydrogenation of glycerol trioleate, castor oil, jojoba oil, olive oil, or very high-oleic sunflower oil | n$_{substrate}$ = 1.6 mmol in water (0.25 mL) and n-heptane (6 mL), T = 30 °C, P = 10 bar H$_2$, t = 60–210 min | ≤45 | - | [79] |

3-Hexyn-1-ol was hydrogenated to 3-hexen-1-ol by Moreno Marrodan and coworkers over the already mentioned lithiated Pd/Dowex$^{TM}$ 50W × 2 IER (see Section 3.1) [67]. Very high values of both conversion (98.5%) and selectivity (>99.8%) were reported in their work.

Four monometallic and six bimetallic catalysts, combining Cu, Ag, Ni, with Pd over the commercial acidic Amberlite$^{TM}$ IR-120 resin in Na$^+$ form, were synthesized and tested for the hydrogenation of 4-nitrophenol and phenylacetylene in Silva et al. (2019). Regarding the monometallic catalysts, Pd, Ag, Ni, or Cu were ion-exchanged with the resin and subsequently reduced with NaBH$_4$. With regards to the bimetallic catalysts, two types are distinguished in their work: Pd-doped catalysts onto which a second metal was introduced by the same method (i.e., ion-exchange followed by NaBH$_4$ reduction) and M-doped catalysts (with M being either Cu, Ag or Ni) onto which Pd was introduced as the second metal [87]. Among the monometallic catalysts, only Pd was active and selective towards ethylbenzene. Interestingly, the simultaneous presence of Ag completely inhibited the catalytic activity of Pd, while the presence of Cu enabled the selective production of styrene over ethylbenzene (i.e., >90% selectivity at high conversion values). On the other hand, Ni-containing catalysts favored ethylbenzene production. The authors attributed the superior behavior of bimetallic Cu catalysts to a geometric effect, rather than to electronic causes.

Sulman et al. (2012) and Nikoshvili et al. (2015) studied the use of hyper-crosslinked polystyrene as support for Pd nanoclusters to selectively hydrogenate C≡C bonds in acetylene alcohols (dimethylethynylcarbinol, dehydrolinalool, and dehydroisophytol) [88,89]. The effect of the solvent nature on the observed activity was investigated, revealing that catalyst activity decreases in the order: alcohols > cyclohexane > water/ethanol mixture > octane ≥ hexane ≥ xylene > toluene > heptane, which apparently correlates to the solvent polarity. According to the authors, neither solvent–substrate interactions or hydrogen solubility in each solvent can explain the reported difference in catalytic activity and selectivity, but the strength of the solvent–catalyst interactions can successfully account for the observed activity pattern [89].

As in the previous section, relevant alkynes hydrogenation reactions can be found in both Scheme 3 and Table 4, which also contain relevant information from the literature.

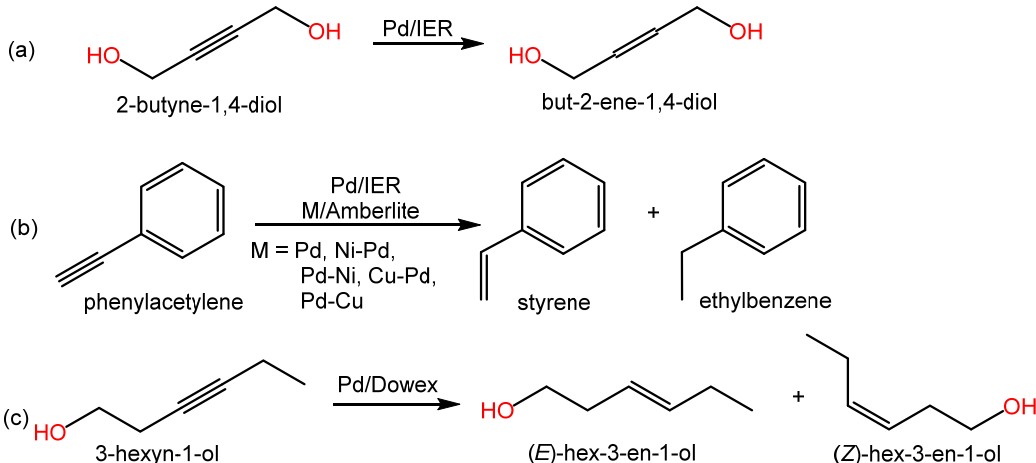

**Scheme 3.** Examples of hydrogenation reactions of alkenes over metal-doped IER in the revised literature.

**Table 4.** Hydrogenation reactions of alkynes over metal-doped IERs in the literature.

| Resin | | | Metal | Reduction Protocol | Tested Reaction | Reaction Conditions | Conversion (%) | Selectivity (%) | Ref. |
|---|---|---|---|---|---|---|---|---|---|
| Type | Polymer | Functional Group | | | | | | | |
| Gel-type | Glycidyl methacrylate (GMA), ST, DEGDMA | –NH–CH$_2$CH$_2$–NH$_2$ | Pd (0.125–0.50 wt.%) | THF:H$_2$O solution of N$_2$H$_4 \times$ H$_2$O | Hydrogenation of 2-butyne-1,4-diol to 2-butene-1,4-diol | $C_{substrate}$ = 0.052 M in THF; T = 22 °C; atmospheric pressure | 90 | 91.3–94.7 | [75] |
| Gel-type | GMA, ST, DEGDMA | –NH–CH$_2$CH$_2$–NH$_2$ | Pd (0.125–0.50 wt.%) | THF:H$_2$O solution of N$_2$H$_4 \times$ H$_2$O | Hydrogenation of phenylacetylene to styrene | $C_{substrate}$ = 0.052 M in THF; T = 22 °C; atmospheric pressure | 90 | 91.4–93.6 | [75] |
| Gel-type | Dowex$^{TM}$ 50W × 2, ST-DVB | –SO$_3$(Li) | Pd (1.3 wt.%) | none | Hydrogenation of 3-hexyn-1-ol to 3-hexen-1-ol | $C_{substrate}$ = 0.17 M in MeOH, room temperature, P = 0.8 bar H$_2$, t = 20 min | 98.5 | >99.8 | [67] |
| Hyper-crosslinked | Macronet$^{TM}$ MN270, ST-DVB | none | Pd (0.1–5 wt.%) | Saturation with H$_2$ for 1 h | Hydrogenation of 2-methyl-3-butyn-2-ol to 2-methyl-3-butene-2-ol | V = 30 mL toluene, T = 90 °C, atmospheric pressure | 100 | 95.3–98.5 | [88] |
| Hyper-crosslinked | Macronet$^{TM}$ MN270, ST-DVB | none | Pd (0.1–5 wt.%) | Saturation with H$_2$ for 1 h | Hydrogenation of 3,7-dimethyloct-6-en-1-yn-3-ol to 3,7-dimethyl-1,6-octadien-3-ol | V = 30 mL toluene, T = 90 °C, atmospheric pressure | 100 | 96.5–98.5 | [88] |
| Hyper-crosslinked | Macronet$^{TM}$ MN270, ST-DVB | none | Pd (0.1–5 wt.%) | Saturation with H$_2$ for 1 h | Hydrogenation of 3,7,11,15-tetramethylhexadec-1-yn-3-ol to 3,7,11,15-tetramethyl-1-hexadecene-3-ol | V = 30 mL toluene, T = 90 °C, atmospheric pressure | 100 | 95.2–97.5 | [88] |
| Hyper-crosslinked | Macronet$^{TM}$ MN270, ST-DVB | none | Pd (0.2 wt.%) | H$_2$ at 300 °C for 2 h | Hydrogenation of 2-methyl-3-butyn-2-ol to 2-methyl-3-butene-2-ol | T = 60 °C, P = 3 bar H$_2$, 1500 rpm in EtOH or toluene as solvents | 95 | 93.2–99.6 | [89] |
| Hyper-crosslinked | Dowex$^{TM}$ OPTIPORE, ST | none | Pd (0.5 wt.%) | H$_2$ at 300 °C for 2 h | | | 95 | 83.0–93.5 | [89] |
| Gel-type | AmberLite$^{TM}$ IRC120, ST-DVB | –SO$_3$(Na) | Pd (0.08 wt.%) | NaBH$_4$ | Hydrogenation of phenylacetylene to styrene | $n_{substrate}$ = 0.5 mmol in 1 mL EtOH, T = 60 °C, P = 4 bar H$_2$, t = 1.1 h | >99 | 0 | [87] |
| | | | Pd (0.09 wt.%), Ag (0.07 wt.%) | NaBH$_4$ | | $n_{substrate}$ = 0.5 mmol in 1 mL EtOH, T = 60 °C, P = 4 bar H$_2$, t = 1.1 h | 12 | - | [87] |
| | | | Ni (0.08 wt.%), Pd (0.10 wt.%) | NaBH$_4$ | | $n_{substrate}$ = 0.5 mmol in 1 mL EtOH, T = 60 °C, P = 4 bar H$_2$, t = 1.1 h | >99 | 7 | [87] |
| | | | Pd (0.06 wt.%), Ni (0.05 wt.%) | NaBH$_4$ | | $n_{substrate}$ = 0.5 mmol in 1 mL EtOH, T = 60 °C, P = 4 bar H$_2$, t = 1.1 h | >99 | 0 | [87] |
| | | | Cu (0.084 wt.%), Pd (0.06 wt.%) | NaBH$_4$ | | $n_{substrate}$ = 0.5 mmol in 1 mL EtOH, T = 60 °C, P = 4 bar H$_2$, t = 1.1 h | 67 | 91 | [87] |
| | | | Pd (0.06 wt.%), Cu (0.10 wt.%) | NaBH$_4$ | | $n_{substrate}$ = 0.5 mmol in 1 mL EtOH, T = 60 °C, P = 4 bar H$_2$, t = 1.1 h | 83 | 93 | [87] |

Only gel-type IERs and hyper-crosslinked unfunctionalized polymers have been used as support for alkynes hydrogenations in the revised literature, with Pd being the most used metal. Interestingly, Ag, Ni, nor Cu showed any catalytic activity when used as monometallic catalysts supported on an Amberlite[TM] resin for the hydrogenation of phenylacetylene [87]. Again, different reduction protocols have been reported with no clear data elucidating the most favorable ones towards the hydrogenation of alkynes.

### 3.3. Hydrogenation of Carbonyl Compounds

Pd-doped, highly lipophilic gel-type IERs, based on styrene or dodecyl methacrylate, were used as catalysts for the hydrogenation of 2-ethylanthraquinone to 2-ethylanthrahydroquinone, which involves the consecutive hydrogenations of C=O and C=C bonds, in both Biffis et al. (2002) and Bombi et al. (2003). Reported selectivity for the tested ST-DVB supported catalyst was 50 and 65%, depending on the reducing step procedure ($H_2$ or $NaBH_4$, respectively), and catalytic activity was clearly lower than a benchmark industrial catalyst (Pd/silicoaluminate). On the other hand, while activity was also low, the resins based on dodecyl methacrylate reached selectivity values as high as 98%, which were slightly above the commercial alternative at the time [45,90].

Hydrogenation of citral (3,7-dimethyl-2,6-octadienal) to geraniol (*trans*-3,7-dimethyl-2,6-octadienol) and nerol (*cis*-3,7-dimethyl-2,6-octadienol) was accomplished by Centomo and coworkers over Pt (0.43–2.5 wt.%) catalysts supported on two sets of functional gel-type resins synthesized by the authors, as well as a commercial 4-vinylpyridine (VP) polymer crosslinked with 2 mol% of DVB [50]. The first set of resins contained *N,N*-dimethyl-2-aminoethylmethacrylate (DMAEMA), cyanoethyl-acrylate (CEA) or methacrylic acid (MAA) as the functional monomer, and DVB as the crosslinker, while the second one contained either CEA or MAA, as the functional monomer, together with *N,N*-dimethylacrylamide (DMAA) and DVB. In addition, two different approaches to incorporate Pt into the resins were adopted: one based on the impregnation of the resin with mesitylene solutions of colloidal platinum, and another one involving immobilization of Pt precursors in pre-swollen resins, followed by chemical reduction. In addition, some of the tested resins were further modified by incorporation of a second metal, acting as Lewis acids (i.e., the introduction of 6.2–15.2 wt.% $Fe^{II}$, $Co^{II}$, or $Zn^{II}$ in Pt-containing resins with –COOH groups). The authors state that improved selectivity can be achieved thanks to the presence of a more electropositive metal than the noble one because it would release electronic density to the noble active metal, which ultimately hinders C=C hydrogenation and favors that of C=O bonds. Reported conversions span from relatively low values (20–30%) to ≥80–90%, depending on the tested catalyst, with selectivity values reaching values above 90%, particularly with the bimetallic catalysts.

The C=O bond in methyl benzoylformate (methyl 2-oxo-2-phenylacetate) and 2,2,2-trifluoroacetophenone (2,2,2-trifluoro-1-phenylethan-1-one) was hydrogenated by Moreno Marrodan and coworkers over a lithiated Pd/Dowex[TM] 50W × 2 IER, yielding selectivity values over 99.5% at acceptably high conversion levels (*ca.* 90% and 62%) when no pre-reduction of the $Pd^{II}$ species supported on the IER was conducted [67].

Another noteworthy application of IERs in the field of carbonyl hydrogenation is that of Barbaro and associates, using Dowex[TM] 50W × 2 resin as support for immobilizing an Ir complex. Their catalyst was used for the hydrogenation reactions of a number of substrates, including several imines, the α-keto ester dihydro-4,4-dimethyl-2,3-furandione, and terpenes like (*R*)-carvone, with similar results to homogeneous catalysts [91]. On the other hand, Pt-doped hyper-crosslinked polystyrene modified with cinchonidine was used as the catalyst for enantioselective hydrogenation of the C=O bond in ethylpyruvate into (*R*)-ethyllactate and (*S*)-ethyllactate in Bykov et al. (2009) [92]. The authors found that enantioselectivity of activated ketone hydrogenation with platinated hyper-crosslinked polystyrene depends on the solvent, reducing agent, temperature, substrate, catalyst, and the modifier concentrations.

Examples of C=O hydrogenation reactions and relevant data are shown in Scheme 4 and Table 5.

**Table 5.** Hydrogenation reactions of carbonyl groups over metal-doped IERs in the literature.

| Resin | | | Metal | Reduction Protocol | Tested Reaction | Reaction Conditions | Conversion (%) | Selectivity (%) | Ref. |
|---|---|---|---|---|---|---|---|---|---|
| Type | Polymer | Functional Group | | | | | | | |
| Gel-type | DMA, MMA, SEMA, EDMA | –$SO_3H$ | Pd (1 wt.%) | $NaBH_4$ | Hydrogenation of 2-ethylanthraquinone | T = 20 °C, P = 100 kPa, t = 1800 s | - | 93 | [45] |
| | | | | $H_2$ | | T = 20 °C, P = 100 kPa, t = 5100 s | - | 94 | [45] |
| Gel-type | DMA, VP, EDMA | –$CH_2CHC_5H_4N$ | Pd (1 wt.%) | $NaBH_4$ | | T = 20 °C, P = 100 kPa, t = 1800 s | - | 93 | [45] |
| | | | | $H_2$ | | T = 20 °C, P = 100 kPa, t = 3600 s | - | 83 | [45] |
| Gel-type | DMA, MAA, EDMA | –COOH | Pd (1 wt.%) | $NaBH_4$ | | T = 20 °C, P = 100 kPa, t = 2400 s | - | 96 | [45] |
| | | | | $H_2$ | | T = 20 °C, P = 100 kPa, t = 16,200 s | - | 98 | [45] |
| Gel-type | ST, SEMA, DVB | –$SO_3H$ | Pd (1 wt.%) | $NaBH_4$ | | T = 20 °C, P = 100 kPa, t = 8400 s | - | 65 | [45] |
| | | | | $H_2$ | | T = 20 °C, P = 100 kPa, t = 8400 s | - | 50 | [45] |
| Gel-type | *N,N*-dimethyl-2-aminoethyl-methacrylate (DMAEMA), DVB | –C(O)O $CH_2CH_2N(CH_3)_2$ | Pt (0.9 wt.) | In situ at 70 °C for 1 h | Hydrogenation of 3,7-dimethyl-2,6-octadienal | T = 60 °C, atmospheric pressure under $H_2$ flow, 25 mL EtOH | 20–30 | ≤15 | [50] |
| Gel-type | Cyanoethyl-acrylate (CEA), DVB | –CN | Pt (1.0 wt.) | | | | ≥80–90 | 46–47 | [50] |
| Gel-type | MAA, DVB | –COOH | Pt (0.8 wt.) | | | | ≥80–90 | 46–47 | [50] |
| Gel-type | VP, DVB | –4-$C_5H_4N$ | Pt (1.0 wt.) | | | | ≤20–30 | 54 | [50] |
| Gel-type | MAA, DVB | –COOH | Pt (0.71 wt.), Fe (8.2 wt.%) | | | | 20–30 | ≥80–90 | [50] |
| | | | Pt (0.89 wt.), Co (14.1 wt.%) | | | | 80–90 | ≤80 | [50] |
| | | | Pt (0.43 wt.), Zn (15.2 wt.%) | | | | 20–30 | ≥80–90 | [50] |
| Gel-type | VP, DVB | –4-$C_5H_4N$ | Pt (2.5 wt.), Co (6.2 wt.%) | | | | 80–90 | ≥80–90 | [50] |
| Gel-type | Dowex™ 50W × 2, ST-DVB | –$SO_3$(Li) | Pd (1.3 wt.%) | In situ | Hydrogenation of methyl 2-oxo-2-phenylacetate | $C_{substrate}$ = 0.17 M in MeOH, room temperature, P = 0.8 bar $H_2$, t = 20 min. | 89.8 | >99.5 | [67] |
| | | | Pd (1.2 wt.%) | 2 bar $H_2$ in MeOH | | | 32.4 | >99.5 | [67] |
| | | –$SO_3H$ | Pd (1.5 wt.%) | In situ | | | 52.6 | >99.5 | [67] |
| | | | Pd (1.3 wt.%) | $H_2$ flow | | | 36.9 | >99.5 | [67] |
| Gel-type | Dowex™ 50W × 2, ST-DVB | –$SO_3$(Li) | Pd (1.3 wt.%) | In situ | Hydrogenation of 2,2,2-trifluoro-1-phenylethan-1-one | $C_{substrate}$ = 0.17 M in MeOH, room temperature, P = 0.8 bar $H_2$, t = 20 min. | 65.4 | >99.8 | [67] |

(a) methyl 2-oxo-2-phenylacetate → Pd/Dowex → methyl 2-hydroxy-2-phenylacetate

(b) 2,2,2-trifluoro-1-phenylethan-1-one → Pd/Dowex → 2,2,2-trifluoro-1-phenylethan-1-ol

(c) (E/Z)-3,7-dimethyl-2,6-octadienal (citral) → M/IER, M = Pt, Pt-Fe, Pt-Co, Pt-Zn → (E)-3,7-dimethyl-2,6-octadienol (nerol) + (Z)-3,7-dimethyl-2,6-octadienol (geraniol)

(d) 2-ethylanthraquinone → Pd/IER → 2-ethylanthrahydroquinone

**Scheme 4.** Examples of hydrogenation reactions of carbonyl groups over metal-doped IERs in the revised literature.

Interestingly, Pd/IER catalysts have been mostly used in the revised literature to hydrogenate C=O bonds, despite Pd being more often related to C=C hydrogenations [86]. On the contrary, even though Ru-based catalysts are typically considered among the most favored ones for the hydrogenation of C=O bonds [93], scarce references report on the use of Ru/IER for this type of reaction. To date, the only references available correspond to works conducted by Barbaro and coworkers (2014, 2016), which will be further addressed in Section 3.7, since they are focused on multistep processes involving sequential C=O hydrogenation, followed by dehydration or lactonization reactions to obtain biomass-derived compounds [94,95]. Consequently, further investigation into the potential use of Ru-doped IER for C=O hydrogenations is advised.

### 3.4. Hydrogenation of Substituted Arenes

In the cited work by Moreno Marrodan et al. (2012), hydrogenation of methyl benzoylformate and 2,2,2-trifluoroacetophenone actually constitute examples of C=O hydrogenation reactions in substituted arenes [67]. Further on, in Moreno Marrodan et al. (2015), the number of substrates belonging to the family of substituted arenes was extended, and selective C=C hydrogenation reactions over a 1.3 wt.% Rh catalyst (3 nm) supported on Dowex$^{TM}$ 50W × 2, either in lithiated or protonated form, was investigated [96]. In their study, the effect of the bead size (i.e., beads of 38–75 μm and of 150–300 μm were used as support), the activation protocol to reduce Rh$^I$ to Rh$^0$ (including no pre-reduction step, reduction with H$_2$, and chemical reduction with NaBH$_4$), and catalyst recyclability was evaluated. Results showed that as-prepared lithiated Rh-doped IERs are efficient C=C hydrogenation catalysts under undemanding conditions (room temperature and 1 bar H$_2$), yielding conversion and selectivity values as high as 100%, depending on the substrate.

In the already mentioned reference by Silva and collaborators [87], besides the hydrogenation of phenylacetylene to styrene (already discussed in Section 3.2), further hydrogenation to ethylbenzene was also studied. Over 99% conversion is reported for the

monometallic Pd-doped IER and the bimetallic catalysts containing Ni. Contrarily, Ag- and Cu-containing catalysts exhibited virtually nil catalytic activity. Seki and coworkers also studied the hydrogenation of benzaldehyde in supercritical $CO_2$ over a Pd catalyst supported on Amberlyst[TM] 15 [83]. Modest results were reported in terms of both conversion (16%) and selectivity (33%).

Regarding the use of Pd-containing, hyper-crosslinked ST, the kinetics of the gas-phase phenol hydrogenation to cyclohexanone were evaluated by Sulman and coworkers [97]. Selectivity greater than 95% is reported at 99% conversion, using Macronet[TM] MN270 as the Pd support.

Relevant literature examples regarding the hydrogenation of substituted arenes using IERs and related data are compiled in Scheme 5 and Table 6.

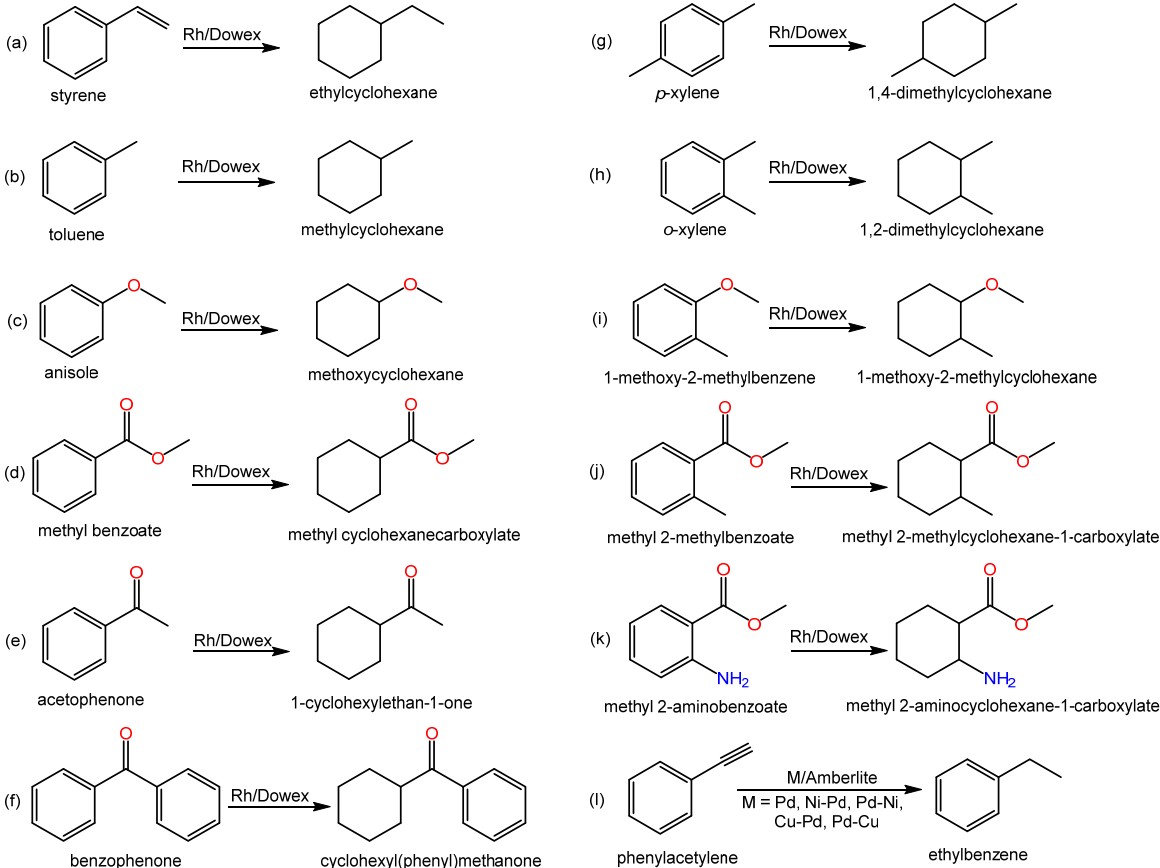

**Scheme 5.** Examples of hydrogenation reactions of substituted arenes over metal-doped IERs in the revised literature.

**Table 6.** Hydrogenation reactions of substituted arenes over metal-doped IERs in the literature.

| Resin | | | Metal | Reduction Protocol | Tested reaction | Reaction Conditions | Conversion (%) | Selectivity (%) | Ref. |
|---|---|---|---|---|---|---|---|---|---|
| Type | Polymer | Functional Group | | | | | | | |
| Macroreticular | Amberlyst™ 15, ST-DVB | –$SO_3H$ | Pd (1 wt.%) | 100 °C for 1 h under a $H_2/N_2$ flow | Hydrogenation of benzaldehyde | Supercritical $CO_2$, T = 60 °C, total P = 160 bar, benzaldehyde:$H_2$ = 1:2 | 16 | 33 | [83] |
| Gel-type | Dowex™ 50W × 2, ST-DVB | –$SO_3$(Li) | Rh (1.3 wt.%) | In situ | Hydrogenation of styrene to ethylcyclohexane | $C_{substrate}$ = 0.16 M in MeOH, room temperature, P = 1 bar $H_2$, t = 380 min | 99.8 | 89.9 | [96] |
| Gel-type | Dowex™ 50W × 2, ST-DVB | –$SO_3$(Li) | Rh (1.3 wt.%) | In situ | Hydrogenation of toluene to methylcyclohexane | $C_{substrate}$ = 0.16 M in MeOH, room temperature, P = 1 bar $H_2$, t = 345 min | 95 | 100 | [96] |
| Gel-type | Dowex™ 50W × 2, ST-DVB | –$SO_3$(Li) | Rh (1.3 wt.%) | In situ | Hydrogenation of anisole to methoxycyclohexane | $C_{substrate}$ = 0.16 M in MeOH, room temperature, P = 10 bar $H_2$, t = 240 min | 98 | 63.7 | [96] |
| | | | | | | $C_{substrate}$ = 0.16 M in MeOH, T = 60 °C, P = 15 bar $H_2$, t = 240 min | 100 | 71.7 | |
| Gel-type | Dowex™ 50W × 2, ST-DVB | –$SO_3$(Li) | Rh (1.3 wt.%) | In situ | Hydrogenation of methyl benzoate to methyl cyclohexanecarboxylate | $C_{substrate}$ = 0.16 M in MeOH, room temperature, P = 10 bar $H_2$, t = 240 min | 73.8 | 97.2 | [96] |
| | | | | | | $C_{substrate}$ = 0.16 M in MeOH, T = 60 °C, P = 10 bar $H_2$, t = 240 min | 100 | 100 | [96] |
| Gel-type | Dowex™ 50W × 2, ST-DVB | –$SO_3$(Li) | Rh (1.3 wt.%) | In situ | Hydrogenation of acetophenone to 1-cyclohexylethan-1-one | $C_{substrate}$ = 0.16 M in MeOH, room temperature, P = 10 bar $H_2$, t = 240 min | 99.3 | 7.2 | [96] |
| Gel-type | Dowex™ 50W × 2, ST-DVB | –$SO_3$(Li) | Rh (1.3 wt.%) | In situ | Hydrogenation of benzophenone to cyclohexyl(phenyl)methanone | $C_{substrate}$ = 0.16 M in MeOH, room temperature, P = 10 bar $H_2$, t = 240 min | 7.5 | 17.3 | [96] |
| Gel-type | Dowex™ 50W × 2, ST-DVB | –$SO_3$(Li) | Rh (1.3 wt.%) | In situ | Hydrogenation of p-xylene to 1,4-dimethylcyclohexane | $C_{substrate}$ = 0.16 M in MeOH, room temperature, P = 10 bar $H_2$, t = 240 min | 100 | 100 | [96] |
| Gel-type | Dowex™ 50W × 2, ST-DVB | –$SO_3$(Li) | Rh (1.3 wt.%) | In situ | Hydrogenation of o-xylene to 1,2-dimethylcyclohexane | $C_{substrate}$ = 0.16 M in MeOH, room temperature, P = 10 bar $H_2$, t = 240 min | 63.2 | 100 | [96] |
| | | | | | | $C_{substrate}$ = 0.16 M in MeOH, T = 60 °C, P = 15 bar $H_2$, t = 240 min | 100 | 100 | [96] |

**Table 6.** *Cont.*

| Resin | | | Metal | Reduction Protocol | Tested reaction | Reaction Conditions | Conversion (%) | Selectivity (%) | Ref. |
|---|---|---|---|---|---|---|---|---|---|
| Type | Polymer | Functional Group | | | | | | | |
| Gel-type | Dowex$^{TM}$ 50W × 2, ST-DVB | –SO$_3$(Li) | Rh (1.3 wt.%) | In situ | Hydrogenation of 1-methoxy-2-methylbenzene to 1-methoxy-2-methylcyclohexane | C$_{substrate}$ = 0.16 M in MeOH, T = 40 °C, P = 15 bar H$_2$, t = 240 min | 100 | 70.9 | [96] |
| Gel-type | Dowex$^{TM}$ 50W × 2, ST-DVB | –SO$_3$(Li) | Rh (1.3 wt.%) | In situ | Hydrogenation of methyl 2-methylbenzoate to methyl 2-methylcyclohexane-1-carboxylate | C$_{substrate}$ = 0.16 M in MeOH, room temperature, P = 10 bar H$_2$, t = 240 min | 31.7 | 85.2 | [96] |
| Gel-type | Dowex$^{TM}$ 50W × 2, ST-DVB | –SO$_3$(Li) | Rh (1.3 wt.%) | In situ | Hydrogenation of methyl 2-aminobenzoate to methyl 2-aminocyclohexane-1-carboxylate | C$_{substrate}$ = 0.16 M in MeOH, T = 60 °C, P = 40 bar H$_2$, t = 4320 min | 76.2 | 37.1 | [96] |
| Gel-type | AmberLite$^{TM}$ IRC120, ST-DVB | –SO3(Na) | Pd (0.08 wt.%) | NaBH$_4$ | Hydrogenation of phenylacetylene to ethylbenzene | 0.5 mmol substrate, 1 mL EtOH, T = 60 °C, P = 4 bar H$_2$, t = 1.1 h | >99 | >99 | [87] |
| | | | Ni (0.08 wt.%), Pd (0.10 wt.%) | NaBH$_4$ | | 0.5 mmol substrate, 1 mL EtOH, T = 60 °C, P = 4 bar H$_2$, t = 1.9 h | >99 | 93 | [87] |
| | | | Pd (0.06 wt.%), Ni (0.05 wt.%) | NaBH$_4$ | | 0.5 mmol substrate, 1 mL EtOH, T = 60 °C, P = 4 bar H$_2$, t = 0.6 h | >99 | >99 | [87] |
| | | | Cu (0.084 wt.%), Pd (0.06 wt.%) | NaBH$_4$ | | 0.5 mmol substrate, 1 mL EtOH, T = 60 °C, P = 4 bar H$_2$, t = 13 h | 67 | 6 | [87] |
| | | | Pd (0.06 wt.%), Cu (0.10 wt.%) | NaBH$_4$ | | 0.5 mmol substrate, 1 mL EtOH, T = 60 °C, P = 4 bar H$_2$, t = 9 h | 83 | 7 | [87] |

### 3.5. Hydrogenation of Nitroaromatic Compounds

Regarding the hydrogenation of nitroaromatics to their corresponding amines, some mentions can be retrieved from Corain et al. (2003, 2010) regarding Pd-doped gel-type IER, but with no specific data besides references contained therein [25,27]. In Gelbard (2005), the nitrobenzene reduction to phenylhydroxylamine, with hydrazine as the reducing agent, over Pt supported on chlorinated Amberlite^TM IRA-410 is reported to achieve a 95% yield [4].

In Silva et al. (2019) (see Sections 3.2 and 3.4), the hydrogenation of 4-nitrophenol to 4-aminophenol was evaluated with the mentioned mono- and bimetallic catalysts supported on Amberlite^TM IR-120 resin. In general, bimetallic catalysts showed faster reaction rates than monometallic ones, except for monometallic Cu, which was the most active one. This is explained in terms of increased catalytically active sites and better availability [87].

### 3.6. Hydrogenation of Nitrates

The aqueous catalytic reduction of nitrates to molecular nitrogen with hydrogen was studied by Gašparovičová and coworkers (2006, 2007) over bimetallic Pd-Cu catalysts (4 wt.% Pd, 1 wt.% Cu) supported on Dowex^TM $1 \times 4$ in the chloride form [98,99]. Dowex^TM $1 \times 4$ is a gel-type ST-DVB resin functionalized with $-N(CH_3)_3^+$ groups. In the first study, five reduction protocols were adopted (C1: $NaBH_4$ in ethanol; C2: $NaBH_4$ in water; C3: $0.1$ MPa $H_2$, $Na_2CO_3$ in $H_2O$; C4: $0.5$ MPa $H_2$, $Na_2CO_3$ in $H_2O$; and C5: $0.5$ MPa $H_2$, $Na_2CO_3$ in methanol) and results were compared. Distinct Pd nanoparticles sizes were observed for each reduction protocol, with C1 being undetectable (which suggests either an amorphous metal phase or with very small particles), C2 presenting the largest particles (i.e., 6.7 nm), and C5 the smallest size (i.e., 2.9 nm). As for the catalytic outcome, the highest selectivity to $N_2$ was achieved by C1 (96%) at 71% nitrates conversion but with the largest Cu leaching detected. Overall, the best catalyst, according to the authors, was C5 since it yielded 73% conversion, with 77% selectivity towards $N_2$, and the lowest Cu leaching. For all catalysts, Pd leaching was below detection limits [98].

In Gašparovičová et al. (2007), the Pd-Cu/Dowex^TM catalyst was compared to Pd-Cu/Al$_2$O$_3$. Two reduction protocols were applied to either type of catalyst. For the resin-based catalyst, the protocols were as follows: $H_2$ in 2.5% water solution of $Na_2CO_3$ for 1 h at 25 °C and 0.5 MPa (catalyst 1), or 0.066 M solution of $NaBH_4$ in ethanol for 1 h at ambient temperature under occasional stirring (catalyst 2). Interestingly, catalyst 2 presented a very homogeneous distribution of Pd and Cu, with Pd- and Cu-containing phases too small to be detected by XRPD, or amorphous, whereas Pd and Cu were mainly located in a shallow external layer of catalyst 1, with Pd nanoparticles of 3.8 nm. This suggests a clear effect of the reduction protocol in the metal distribution within the resin beads. Noticeably, expected byproducts, namely nitrites (produced by partial reduction) and ammonia (over reduction), were almost avoided with resin-based catalysts (i.e., selectivity towards $N_2$ was 93.8% at 44% conversion with catalyst 2). On the other hand, Pd-Cu/Al$_2$O$_3$ catalysts were more active, yet less selective to $N_2$ (below 60%) than resin catalysts. Furthermore, while Pd-Cu/Al$_2$O$_3$ catalysts showed some leaching of Cu (no leaching of Pd was detected), Cu and Pd in solution were below detection limits for resin-based catalysts [99].

Another Pd-Cu catalyst was investigated by Mendow and coworkers for the same reaction system, with the emphasis being made on the performance of two processes for nitrates removal [100]. In their work, Mendow used a macroporous, ST-DVB anion exchange resin (i.e., Diaion^TM WA30), containing tertiary amine functional groups. Pd incorporation in the resin was accomplished by contact with $PdCl_2$ solution dissolved in NaCl 0.01 M and HCl 0.01 M while bubbling $N_2$. After reduction of the $Pd^{II}$ to $Pd^0$ with 35 wt.% hydrazine solution, Cu was incorporated by means of the controlled surface reduction method. In this procedure, the reduced Pd/WA30 catalyst was suspended in water and $H_2$ was bubbled for 2 h; then, a $CuNO_3·3H_2O$ solution was added while maintaining the hydrogen bubbling for 2 h. After several additional steps involving filtration, reduction, and washing operations, the final metal contents were 2 wt.% Pd

and 0.5 wt.% Cu, as verified by energy-dispersive X-ray fluorescence (XRF). Depending on the operating conditions, conversion, and selectivity to $N_2$ values as high as 100% are reported [100].

The simultaneous catalytic reduction of nitrate ions and reductive dehalogenation of organochlorinated pollutants from water was studied by Bradu and associates, who prepared bimetallic Pd-Cu catalysts over Purolite® A520E, which is a strong base anion exchange resin of the macroreticular type, presenting quaternary ammonium functional groups on an ST-DVB polymer matrix [101]. Four catalysts were prepared and compared: (i) a monometallic Pd (2.01 wt.%) over A520E was prepared by ion exchange; (ii) a bimetallic Pd-Cu (1.97 wt.% and 0.45 wt.%, respectively) catalyst sample was prepared by simultaneous ion-exchange of both metal salts with the resin; (iii) a second bimetallic Pd-Cu (1.98 wt.%, 0.48 wt.%) catalyst was prepared by consecutive stages of ion-exchange of each metal salt with the resin, with Pd being exchanged in the first place; and (iv), finally, a third bimetallic Pd-Cu (2.01 wt.%, 0.49 wt.%) catalyst was prepared by Pd ion-exchange, followed by Cu deposition by controlled surface reaction. The latter catalyst sample was able to achieve selective nitrate reduction, with a 95% conversion and 92% selectivity towards $N_2$, with almost quantitative hydrodechlorination of 4-chlorophenol [101].

### 3.7. One-Pot Multistep Reaction Processes Involving Hydrogenation

Regarding multistep processes, several relevant examples can be found in the literature, signaling that this specific field is of great interest since it deals directly with the integration and optimization of processes enabling the one-pot synthesis of valuable chemicals. For instance, the simultaneous hydrogenation and isomerization of diisobutylenes (2,4,4-trimethylpent-1-ene, TMP1, and 2,4,4-trimethylpent-2-ene, TMP2) over a Pd-doped IER was investigated by Talwalkar and coworkers [102]. Interestingly, the commercial Amberlyst™ CH28 bifunctional catalyst, which contains Pd nanoparticles and –$SO_3H$ functional groups in a macroreticular matrix, was found to favor isomerization of TMP2 to TMP1, which is more prone to hydrogenation towards isooctane than TMP2 due to the terminal position of the double bond. From this, it follows that bifunctionalization of a resin with acid and metallic groups can offer selectivity changes in isomerization reactions, making the subsequent hydrogenation step more profitable. The effects of operating parameters (i.e., temperature, catalyst loading, hydrogen pressure, impurities associated with TMPs, and catalyst reusability) were evaluated and a kinetic model was proposed.

Seki et al. (2007, 2008) reported on the one-pot synthesis of 2-ethylhexanal from crotonaldehyde, which involves hydrogenation and aldol condensation in supercritical $CO_2$, over a bifunctional acidic resin-supported palladium catalyst [83,103]. They incorporated Pd (1 wt.%) onto Amberlyst™ 15 and obtained 13–98% conversions and ~0–67% selectivity values, depending on the assayed conditions.

The one-pot synthesis of a potential analgesic over Pd/Amberlyst™ 15 was studied by Wissler and coworkers. In their work, two reaction steps are involved: dehydration of the starting tertiary alcohol, followed by hydrogenation of the obtained olefin. At the proposed optimized conditions, the reported conversion was 98%, and the selectivity towards the target product was 64%. However, prompt deactivation of the catalyst in the first reuse was observed, which the authors link to possible coke formation on the resin, since they argue that the selected temperature (i.e., 150 °C) is not high enough to promote thermal instability and no Pd leaching was observed [104].

Besides the examples mentioned so far, the specific, yet broad, field of biomass transformation into high-value-added products seems to be particularly suited for using metal-doped IER. For instance, Barbaro et al. (2016) focused on the direct conversion of glucose and xylose to isosorbide and anhydroxylitol, respectively, involving hydrogenation followed by dehydration. Dowex™ 50W × 2 in its $H^+$ form was used as support for Ru, with reported yields of 84.9% for isosorbide and 94.9% for anhydroxylitol at 190 °C with 30 bar $H_2$ in water in batch experiments [94]. Such high yields are attributed to a combined effect of (i) a favorable microporous structure of IER and a narrow size distribution of Ru

nanoparticles, which would enhance hydrogenation selectivity, and, (ii) a proper balance of density and strength of Brønsted acid sites, which would enhance dehydration.

In a different approach, a mixture of Pt/C catalyst and the thermostable, ST-DVB macroreticular resin Amberlyst$^{TM}$ 70 was used for the cellulose conversion into isosorbide, which involves three consecutive reaction steps: hydration, hydrogenation, and dehydration [105]. The same approach was adopted in Galletti et al. (2012) for the synthesis of γ-valerolactone (GVL) from levulinic acid [106].

With respect to GVL production, Au, Pt, Ir, Ni, Cu, Re, Rh, and Ru over different supports have been mentioned as effective catalysts to obtain GVL by hydrogenation of levulinic acid [95,107–111]. In this process, either hydrogenation followed by dehydration or dehydration–hydrogenation reactions take place. As for the use of IERs to produce GVL, Moreno Marrodan and Barbaro (2014) supported Ru (0.87 wt.%) on Dowex$^{TM}$ 50W × 2 by contacting it with RuCl$_3$ and further reduction with NaBH$_4$, with water being used as the solvent in both metalation and reduction steps. Values of 16.2–99.8% conversions in batch runs and 89–100% conversions in continuous runs (residence time = 62–211 s, T = 70 °C, P = 4.8–7.0 bar H$_2$, water) are reported, with selectivity to GVL always above 99%. No significant activity decay was observed after three consecutive runs of recycling the catalyst and no Ru leaching was detected [95].

In another study by Moreno Marrodan and coworkers, metal nanoparticles (Pd, Rh, or Ru) were supported on a perfluorinated, fluorosulfonic acid resin (i.e., Aquivion® PFSA), which is claimed to possess an acidity similar to sulfuric acid, a thermal stability well beyond that of conventional IER, and chemical inertness in aggressive environments. Full selectivity at high conversion levels is reported for the conversion of (+)-citronellal to (-)-menthol and levulinic acid to γ-valerolactone (GVL) under mild conditions over Pd-based catalysts [112].

Another prominent reaction system within this field is the synthesis of methyl isobutyl ketone (MIBK) from acetone, involving consecutive steps of condensation, dehydration, and hydrogenation (e.g., [77,113–119]). The commercial Pd-doped Amberlyst$^{TM}$ CH28 was pinpointed as an effective catalyst to carry out this synthesis. Depending on the operating conditions, reported acetone conversion values span from 25 to >99%, with selectivity in the range 70–90% [77,113–118]. Amberlyst$^{TM}$ CH43, a Pd-doped strong acid styrenic IER, was reported to provide higher selectivity towards MIBK at high temperatures in a continuous 8 h run [118]. Remarkably, Aquivion® polymers were also used as Pd supports (0.10–1.04 wt.%) for the synthesis of MIBK, showing conversions in the range 11.0–63.3% and selectivity values of 62.8–96.5% in batch runs (t = 2–22 h, T = 120–180 °C, P = 10–25 bar H$_2$) [119].

Several multistep reactions over metal-doped IER are shown in Scheme 6. Table 7 lists some of the information retrieved from the literature covered. As can be seen, all multistep processes found in the literature combine metal sites with acid sites. Both macroreticular and gel-type resins have been used. Furthermore, the commercial Amberlyst$^{TM}$ CH28 needs to be highlighted, since it is a readily available product that has shown good results in different multistep reaction processes.

**Scheme 6.** Multistep reaction processes over metal-doped IERs in the revised literature.

**Table 7.** Multistep reaction processes over metal-doped IERs in the literature.

| Resin | | | Metal | Reduction Protocol | Tested Reaction | Reaction Conditions | Conversion (%) | Selectivity (%) | Ref. |
|---|---|---|---|---|---|---|---|---|---|
| Type | Polymer | Functional Group | | | | | | | |
| Macroreticular | Amberlyst$^{TM}$ CH28, ST-DVB | –SO$_3$H | Pd (0.7 wt.%) | In situ | Simultaneous hydrogenation and isomerization of diisobutylenes | T = 100 °C, P = 40 bar H$_2$, t = 80 min | 100 | 100 | [102] |
| | | | | | | T = 100 °C, P = 30 bar H$_2$, t = 80 min | 81 | 100 | [102] |
| | | | | | | T = 100 °C, P = 30 bar H$_2$, t = 45 min | 54 | 100 | [102] |
| Macroreticular | Amberlyst$^{TM}$ 15, ST-DVB | –SO$_3$H | Pd (1 wt.%) | 100 °C for 1 h under a H$_2$/N$_2$ flow | Production of 2-ethylhexanal from crotonaldehyde | Supercritical CO$_2$, T = 60 °C, total P = 160 bar, crotonaldehyde:H$_2$ = 1:2 | 89 | 47 | [103] |
| | | | | | | Supercritical CO$_2$, T = 60 °C, total P = 160 bar, crotonaldehyde:H$_2$ = 1:3 | 94 | 59 | [103] |
| | | | | | | Supercritical CO$_2$, T = 60 °C, total P = 160 bar, crotonaldehyde:H$_2$ = 1:4 | 98 | 67 | [103] |
| | | | | | | Supercritical CO$_2$, T = 60 °C, total P = 40 bar, crotonaldehyde:H$_2$ = 1:4 | 13 | ~0 | [103] |
| Macroreticular | AmberlystTM 15, ST-DVB | –SO$_3$H | Pd (0.1–3 wt.%) | Either unreduced or prereduced at 140 °C in 10 vol% H$_2$ in Ar | Dehydroxylation of a tramadol derivative | 1.7 mmol substrate in 10 mL EtOH, T = 150 °C, P = 4 bar H$_2$, t = 4 h | >98 | 76 | [104] |
| Gel-type | Dowex$^{TM}$ 50W × 2 | –SO$_3$H | Ru (0.2 wt.%) | NaBH$_4$ | Direct conversion of glucose into isosorbide | Csubstrate = 0.1 M in water, T = 190 °C, P = 30 bar H$_2$, t = 48 h | 100 | 84.9 | [94] |
| | | | | | | Csubstrate = 0.1 M in water, T = 120 °C, P = 30 bar H$_2$, t = 7 h | 100 | 0 | [94] |
| Gel-type | Dowex$^{TM}$ 50W × 2 | –SO$_3$H | Ru (0.2 wt.%) | NaBH$_4$ | Direct conversion of xylose into anhydroxylitol | Csubstrate = 0.1 M in water, T = 190 °C, P = 30 bar H$_2$, t = 6 h | 100 | 94.9 | [94] |
| | | | | | | Csubstrate = 0.1 M in water, T = 120 °C, P = 30 bar H$_2$, t = 7 h | 99.7 | 0 | [94] |
| Gel-type | DowexTM 50W × 2 | –SO$_3$H | Ru (0.87 wt.%) | NaBH$_4$ | Production of γ-valerolactone from levulinic acid | Csubstrate = 0.43 M in water, T = 70 °C, P = 5 bar H$_2$, t = 7 h | 99.8 | >99 | [95] |
| | | | | | | Csubstrate = 0.43 M in water, T = 70 °C, P = 5 bar H$_2$, t = 4 h | 79.5 | >99 | [95] |
| | | | | | | Csubstrate = 0.43 M in water, T = 50 °C, P = 5 bar H$_2$, t = 4 h | 29.3 | >99 | [95] |
| | | | | | | Csubstrate = 0.43 M in water, T = 70 °C, P = 10 bar H$_2$, t = 4 h | 98.3 | >99 | [95] |
| | | –SO$_3$(Li) | | | | Csubstrate = 0.43 M in water, T = 70 °C, P = 5 bar H$_2$, t = 4 h | 16.2 | >99 | [95] |

**Table 7.** *Cont.*

| Resin | | | Metal | Reduction Protocol | Tested Reaction | Reaction Conditions | Conversion (%) | Selectivity (%) | Ref. |
|---|---|---|---|---|---|---|---|---|---|
| Type | Polymer | Functional Group | | | | | | | |
| Macroreticular | Amberlyst™ CH28, ST-DVB | –SO₃H | Pd (0.7 wt.%) | acetone and H₂ flow at 30 bar for 4 h | Synthesis of methyl isobutyl ketone from acetone | T = 130 °C, P = 30 bar H₂, LHSV = 4 h-1 | 38 | 88.3 | [115] |
| | | | | | | T = 140 °C, P = 30 bar H₂, LHSV = 4 h-1 | 45.5 | 80.8 | [115] |
| | | | | | | T = 150 °C, P = 30 bar H2, LHSV = 4 h-1 | 55.6 | 72.8 | [115] |
| Macroreticular | Purolite CT®275, ST-DVB | –SO₃H | Pd (0.5 wt.%) | H₂ flow | Synthesis of methyl isobutyl ketone from acetone | T = 105–130 °C, P = 30 bar H₂, LHSV = 0.4–2.4 h-1, H₂:acetone = 2:1 to 15:1 | 24–34 | >85 | [116] |
| Macroreticular | Amberlyst™ CH28, ST-DVB | –SO₃H | Pd (0.7 wt.%) | In situ | Synthesis of methyl isobutyl ketone from acetone | T = 100–140 °C, P = 30 bar H₂, t = 12,000–14,000 s | 5–45 | >95 | [77] |
| Macroreticular | Amberlyst™ CH28, ST-DVB | –SO₃H | Pd (0.7 wt.%) | 24 h under H₂ (20 bar) at 100 °C | Synthesis of methyl isobutyl ketone from acetone | Continuous reactor:T = 90–140 °C, P = 15 bar H₂, LHSV = 2.5 h-1 Batch reactor: T = 90–140 °C, P = 14 bar H₂, t = 4 h | 45 | >80–95 | [118] |
| Macroreticular | Amberlyst™ CH43, ST-DVB | –SO₃H | Pd (0.7 wt.%) | 24 h under H₂ (20 bar) at 100 °C | Synthesis of methyl isobutyl ketone from acetone | Continuous reactor:T = 90–140 °C, P = 15 bar H₂, LHSV = 2.5 h-1 Batch reactor: T = 90–140 °C, P = 14 bar H₂, t = 4 h | 45 | >85–95 | [118] |

### 3.8. Other Related Reactions

With regards to the role of IERs in other hydrogenation reactions (or related reaction systems), the direct synthesis of hydrogen peroxide from molecular oxygen and hydrogen stands out as one of the most studied processes. For instance, Sterchele and coworkers successfully prepared bimetallic Pd-Au (Pd: 1.0; Au: 0.25–1.0 wt.%) and Pd-Pt (Pd: 1.0; Pt: 0.1–1.0 wt.%) catalysts over a commercial, ST-DVB macroreticular resin, Lewatit® K2621, by simple ion-exchange in water and reduction with aqueous formaldehyde, which they tested in the direct synthesis of hydrogen peroxide [58]. They found that the presence of small amounts of either Pt or Au in Pd catalysts promoted selectivity while reducing activity in comparison with monometallic Pd catalysts, but that further addition of Pt or Au produced different effects: while activity increases with an increasing amount of Au, a maximum in hydrogen peroxide productivity was observed with 0.5 wt.% Pt addition, but also the lowest selectivity level was achieved. Other relevant examples of works dealing with this reaction system are those by Burato (2006) and Frison (2019), which have already been commented on in previous sections [52,66].

Concerning catalytic hydrodechlorination reactions, Králik et al. (2014) supported Pd and Pt over Dowex[TM] 1 × 4, an ST-DVB gel-type resin, functionalized with trimethylammonium, and the catalysts were tested in the hydrogenation of chloronitrobenzene isomers into the corresponding chloroanilines. The resin was used either in the chloride form or in the $OH^-$ form, and about 1 wt.% metal contents were measured. From ISEC analyses before and after metal deposition, it became evident that lower concentrated polymer domains (i.e., 0.4 nmnm$^{-3}$) decreased with the metal incorporation while increasing higher concentrated domains (0.8 nmnm$^{-3}$) [120]. Regarding the catalytic results, *x*-chloronitrobenzene, *x*-chloroaniline, aniline, *x*-chloronitrosobenzene, and nitrobenzene were detected in the reaction medium after the catalytic tests, with *x*-chloroaniline being the target product in that work (i.e., reduction of the NO$_2$ group, without hydrodechlorination). Significantly higher activity, but lower selectivity to *x*-chloroaniline, was found for the Pd-containing catalysts in the $OH^-$ form than for their chloride counterparts. As indicated by the authors, in previous works by Krátky et al. (2002, 2003), where a Pd-doped, ST-DVB sulfonated IER was compared to Pd/charcoal catalyst and the effect of different solvents was evaluated, it had already been established that condensation reactions occurring in basic environments lowered the selectivity to *x*-chloroaniline [121,122]. All Pt catalysts in Králik et al. (2014) were more active than the Pd ones, which the authors linked to the lower average size of Pt crystallites (2.3 nm) in comparison with the Pd ones (4.5 nm) [120].

Han and coworkers used Amberlite[TM] IRA-900 and IRA-958 (both in Cl$^-$ form) as support for Pd (0.1–11.2 wt.%, 3–5 nm) to catalyze the complete hydrodechlorination of triclosan (5-chloro-2-(2,4-dichlorophenoxy)phenol). Both resins are of the macroreticular type with quaternary amine as the functional group, but IRA-900 is an ST-DVB resin whereas IRA-958 is acrylic. IRA-958 showed better performance than IRA-900, which was attributed to the hydrophobicity, pore size, and surface area of its polymeric matrix, and to larger Pd particle size (4.6 and 3.7 nm for IRA-958 and IRA-900, respectively, but with some larger aggregates of 20–40 nm diameter range observed on IRA-958). Both catalysts were reused multiple times without significant loss in effectivity or leaching [123].

On the other hand, Jadbabaei et al. (2017) used nonionic polymeric resins (that is crosslinked Amberlite[TM] XAD-4 and hyper-crosslinked Purolite® MN200 and MN100, the latter with a small fraction of tertiary amine functional groups) and basic ST-DVB IER (i.e., Amberlite[TM] IRA-910 and IRA-96, with dimethyl-ethanolammonium and tertiary amine functional groups, respectively) as supports for Pd catalysts aiming at the catalytic hydrodechlorination of 4-chlorophenol. A Langmuir-Hinshelwood kinetic model was developed, with surface reaction as the rate-determining step, which suggested an enhancing effect of adsorption on the catalytic reactivity. IRA910 was found to facilitate Cl$^-$ adsorption and promote the production of PdCl$_3$$^-$ and PdCl$_4$$^{2-}$ species, which are inactive for catalytic reduction [124].

Hydrogenolysis of glycerol, which involves chemical bond cleavage with the simultaneous addition of a hydrogen atom to the resulting molecular fragments [125], yields the production of 1,2- and 1,3-propanediol, together with ethylene glycol. Kusunoki et al. (2005), Miyazawa et al. (2006), and Centomo et al. (2013) studied this reaction system in connection to the use of IER. However, physical mixtures of carbon-supported metal catalysts (e.g., Ru/C, Pt/C, Pd/C, Rh/C) with IER were found to be much more selective towards 1,2-propanediol (which is the preferred target product) than tested metal-doped IER [126–128]. Furthermore, in the framework of the hydrogenolysis of glycerol, van Ryneveld and coworkers combined Ru/C catalysts with thermal-resistant IERs, such as Amberlyst[TM] DT and Amberlyst[TM] 70 [129]. Regarding hydrodeoxygenation, a particular type of hydrogenolysis involving oxygen removal from oxygen-containing compounds (which is an interesting reaction system for the valorization of biomass waste streams, e.g., [130]), has been found to be effectively achieved by monometallic Cu/C and bimetallic Mg-Cu/C obtained from the weakly acidic, acrylic IER Diaion[TM] WK11, which was calcinated after ion-exchanging it with the designated precursor salts in Wang et al. (2021, 2022). Particularly relevant features of those works are the reported high metal loadings (>50 wt.%) and small metal particle sizes (<15 nm) [131,132].

## 4. Concluding Remarks and Future Outlook

As can be deducted throughout the work, numerous IER applications exist as catalyst support for metal-catalyzed hydrogenation reactions, making it a wide, vibrant field, with both a solid trajectory and a promising future prospective. Extensive work has already been conducted regarding metal dispersion over IERs, with multiple data being published regarding metal precursor salts, solvents, and conditions for the ion exchange between the resin functional groups and the metal ions. Likewise, and strongly dependent on the final application of the metal/IER catalyst, different procedures to activate the metal, that is, to reduce the metal ion to its zero-valence species, can be retrieved from the covered literature, with some works reporting that the reduction protocol would affect the metal distribution within the resin beads.

Both gel-type and macroreticular, basic or acid (yet, with acidic resins being the most widely used ones), IERs have been used for many metal-catalyzed hydrogenation processes, either in the gas and the liquid phase, including hydrogenation of alkenes, alkynes, carbonyls, arenes, nitroaromatics, and nitrates, among others. Of particular interest are the one-pot multistep synthesis processes including hydrogenation, together with hydration, dehydration, condensation, or isomerization reactions, covered in Section 3.7, since they constitute a particular type of application of IERs in the field of hydrogenation, not only as metal supports but also as acid sites-containing catalysts. Such technology allows for the integration and optimization of processes and is encompassed with improved sustainability. In this regard, current trends aiming at the transformation of biomass derivatives into high-value-added chemicals match the potential use of metal-doped IERs as bifunctional catalysts to obtain industrially acceptable target product yields under mild conditions. Even more so when, according to reported data, metal species leaching after extended use of metal-doped IER catalysts does not appear to be a cause for concern.

Another aspect worth mentioning that can be inferred from data provided in Tables 3–7 is the fact that mild operating conditions (that is, temperatures in the range of room temperature to 150 °C and $H_2$ pressures typically below 40 bar) are needed to obtain high yields towards desired products, which contrasts with the more demanding operating conditions usually associated to inorganic supports. This adds to the fact that the metal loading in IERs is usually achieved by simple ion exchange at room temperature, without any thermal-intensive processes such as calcination. Therefore, lower energy-demanding operations are involved in both the manufacturing and the utilization of metal-doped IERs as catalysts when compared to catalysis over inorganic supports. In addition, another potential benefit of using IERs would be related to their effect on critical parameters for the catalytic behavior of these materials, such as metal nanoparticles size as well as

nature and number of defects, since IERs offer distinct microenvironments surrounding the nanoparticles that can boost their effectiveness for a given reaction.

As a matter of fact, the advantages and limitations of using IERs in this field could be summarized as follows: on the one hand, IERs combine hydrophilic and hydrophobic regions, which may provide unique microenvironments to metal nanoparticles and/or promote specific interactions of interest between reactants and the polymer material, ultimately affecting reaction selectivity for a given application. In addition, and depending on the IER composition (e.g., DVB content) and morphology, a molecular sieve action can be in effect when using IERs which also would impact reaction selectivity when steric effects are in place. On the other hand, the main limitation of IERs is their relatively low thermal stability compared to inorganic materials, since only a selected group of IERs can be used at temperatures higher than 150 °C. Furthermore, morphology control and characterization of IERs can be more complex than nanostructured systems such as zeolites, since, for instance, working-state porosity might depend on the properties of the fluid containing the IER.

Following this remark, despite some relationships between IER structures and their performance as solid supports for metal catalysts being established from the covered literature (for instance, metal incorporation into an IER would be related to an increase in its higher concentrated polymer domains), further work is advised in the line of assessing the influence of the swollen polymer morphology and IER functionality, in relation to the used solvent for the metalation step, to secure control of the metal nanoclusters size. Moreover, since most of the revised works used Pd as the catalytically active metal regardless of the hydrogenation type, combination of different metals, and their potential effect on conversion and selectivity, as well as the possible influence of the metal dispersion protocols in obtaining higher yields seems a propitious area to continue exploring. With respect to the metals, the production of mono- or multimetallic catalysts based on non-noble metals (e.g., Ni, Fe, Cu) is still to be fully exploited, given that very few references have been found to date studying their use as catalysts over IERs, which would be well aligned with the present tendency of reducing the use of noble metals, due to their high cost and low availability.

**Author Contributions:** Conceptualization, J.H.B., R.S., E.R., R.B., C.F., M.I. and J.T.; Methodology, J.H.B. and J.T.; Writing—Original Draft Preparation, J.H.B.; Writing—Review and Editing, J.H.B., R.S., E.R., R.B., C.F., M.I. and J.T.; Visualization, J.H.B. and R.S.; Supervision, J.T. All authors have read and agreed to the published version of the manuscript.

**Funding:** This research received no external funding.

**Data Availability Statement:** No new data were created in this review study. Data sharing is not applicable to this article.

**Conflicts of Interest:** The authors declare no conflict of interest.

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
