# Peer review of "Role of Ion-Exchange Resins in Hydrogenation Reactions"

_catalysts, doi:10.3390/catal13030624_

Round 1

Reviewer 1 Report

The review article extensively describes the possible use of ion-exchange resins as supports for metal catalysts in hydrogenation reactions. Notably, the article includes much technical information concerning the synthesis and application of ion-exchange resins. Particularly, I appreciated the use of tables to report all relevant literature case studies and thus summarize meaningful results. The article is reasonably well-written, but the overall readability can still be improved. At any rate, I recommend the publication of the present contribution on the condition that the following comments and suggestions are considered to improve the contribution.

1)    Firstly, I believe it would be important to mention why using ion-exchange resins as catalyst supports is particularly relevant for catalyzed hydrogenation reactions. Can the authors comment shortly on this?

2)    Furthermore, a broad range of substrates has been investigated and thus reported in the contribution. It could be helpful to briefly discuss their importance on a commercial level when relevant. Of course, this should be done only for a few noteworthy examples. As a result, the readability of more monotonous passages will be improved, and more attention will be drawn to the use of ion-exchange resins for catalysis applications.

3)    However, most importantly, the review would probably improve significantly by including a relatively short section summarizing the most important features of ion-exchange resins (partly in Section 1.1) and related advantages and limitations for designated applications, which were not at all discussed. As a result, an outlook on possible progress required by the field will be necessary. To this end, the authors can probably expand Section 1.1 and Conclusions accordingly.

4)    In addition, I have noticed that the contribution focuses more on the synthesis of catalyst materials based on ion-exchange resins than on their application. Particularly, it would be important to mention how the materials compare with established catalysts in more detail for selected examples or at least some general remarks in a dedicated section.

5)    Following up on the previous comment: I advise caution in reporting others’ results and related interpretations. For instance, on page 5 (lines 188 to 201), the activity of Pt-Au catalysts is briefly discussed in the oxidation of glycerol to the acid. In this case, alloying was found to be responsible for the improved activity of PtAu and AuPt catalysts compared to Pt-only materials. However, if not supported by evidence, i.e., phase and microstructure analysis, this conclusion is only speculative and should not receive mention. Otherwise, more information about the characterization of the materials should be reported. Further, on pages 6-8 (lines 267, 268, and 330), the average size of supported Pd nanoparticles is written as 1.34, 2.42, 2.59, and 1.34 nm, respectively. This is usually incorrect because such a level of precision is not expected via electron microscopy analysis. At any rate, I recommend that the authors not risk reporting fault or misleading results but remain vague (e.g., indicate “average sizes between 1-3 nm).

6)    The title of Section 2 is not consistent. Although the title reads: “Ion-exchange resins as catalyst support for hydrogenation reactions”, most examples deal with oxidation and hydroxylation reactions. Besides, the title is way too similar to the one for the following sections. Thus, I would anyway recommend diversifying to improve the overall structure.

7)    It is not uncommon to activate the catalyst material (or pre-catalyst) directly in the reactor, following an established activation protocol, often a reduction with hydrogen at the most suitable temperature, but not under reaction conditions. This could imply an induction period for the catalyst and lead to inaccurate results or troublesome comparisons in the catalytic behavior of different materials. Since the authors have stressed this practice on several occasions while discussing previous literature reports, I would recommend adding at least a note of caution to make the last point clear.

8)    The term “nanoclusters” is inappropriate for describing 30-50 nm metal nanoparticles (page 14, line 465). Metal clusters include a small number of metal atoms arranged in a well-defined geometry. Such assemblies are typically smaller than a few nanometers. More precise information about the nature of metal clusters can be found in the dedicated literature. To avoid mistakes, any mention of the term “nanocluster(s)” (not only on page 14) should better be replaced with “nanoparticle(s)”.

9)    The review only includes 6 references (out of 136) more recent than 2019. Can the authors comment briefly on this?

Reviewer 2 Report

In the manuscript " Role of Ion-Exchange Resins in Hydrogenation Reactions " Tejero et al. presents the role of ion-exchange resins (IERs) as catalysts, or catalysts supports, in hydrogenation reactions. Noble metals, such as Pt, Au, and Pd, and non-noble metals, like Fe and Cu, have been introduced into IERs polymeric backbones by simple ion-exchange of a metal salt precursor with the resin, or by combination of ion-exchange and other protocols, to obtain mono- and bimetallic catalysts supported on IERs. This manuscript is well-organized and carefully written. It can be accepted after minor revision. The comments are presented as follows:

1. “Also, two reviews authored by Osazuwa and Abidin were published recently [13,14], where some works can be found cited therein regarding the use of IERs in different hydrogenations reactions, together with references related to the use of polymeric frameworks different from IERs as supports for metal catalysts (e.g., [15–18])” The latest literature about hydrogen storage technologies should be cited, such as Chao Wan, Liu Zhou, Suman Xu, Biyu Jin, Xin Ge, Xing Qian, Lixin Xu, Fengqiu Chen, Xiaoli Zhan, Yongrong Yang, Dangguo Cheng. Defect engineered mesoporous graphitic carbon nitride modified with AgPd nanoparticles for enhanced photocatalytic hydrogen evolution from formic acid, Chemical Engineering Journal, 2022, 429, 132388. Yu Kai,Ai Zhengrong,Ning Zhiqiang,Hu Wei,Xie Hongwei. Adsorption Process in Aqueous Solution of Silver Nitrate by Ion Exchange Method. Chinese Journal of Rare Metals. 2021,45(11):1352-1358.

2. Both gel-type and macroreticular, basic or acid (yet, with acidic resins being the most widely used ones), IERs have been used for many metal-catalyzed hydrogenation processes, either in the gas and the liquid phase, including hydrogenation of alkenes, alkynes, carbonyls, arenes, nitroaromatics, and nitrates, among others. The comparison table about IERs and other methods should be supplied.
